# CAUSAL INFERENCE USING LLM-GUIDED DISCOVERY

## ABSTRACT

At the core of causal inference lies the challenge of determining reliable causal graphs solely based on observational data. Since the well-known back-door criterion depends on the graph structure, any errors in the estimated graph structure affect the correctness of the estimated causal effects. In this work, to construct a valid back-door adjustment set, we propose to use a topological or causal order among graph nodes, which is easier to get from domain experts. Given a node pair, causal order is easier to elicit from domain experts compared to graph edges since determining the existence of an edge depends extensively on other variables. Interestingly, we observe that the same principle holds for Large Language Models (LLMs) such as GPT-3.5-turbo and GPT-4, motivating an automated method to obtain causal order (and hence causal effect) with LLMs acting as virtual domain experts. To this end, we employ different prompting strategies and contextual cues to propose a robust technique for obtaining causal order from LLMs. Acknowledging LLMs' limitations, we also study possible techniques to integrate LLMs with established causal discovery algorithms, including constraint-based and score-based methods, to enhance their performance. Extensive experiments demonstrate that our approach significantly improves causal ordering accuracy as compared to discovery algorithms, highlighting the potential of LLMs to enhance causal inference across diverse fields.

## 1 INTRODUCTION

Causal inference plays a pivotal role across scientific disciplines, aiding researchers in uncovering fundamental causal relationships and how they affect observed phenomena. For example, causal inference is used to discern the causes of diseases and design effective interventions for diagnosis and treatment in epidemiology (Mahmood et al., 2014), to evaluate policy impact based on observational studies in economics (Imbens & Rubin, 2015), and to understand the effects of pollution on ecosystems in environmental science (Boslaugh, 2023). A key technical question for these studies is estimating the *causal effect* of variables on a specific outcome variable.

Inferring causal effect from observational data, however, is a challenging task because the effect estimate depends critically on the causal graph considered in the analysis. While there has been progress in graph discovery algorithms, especially for specific parametric settings (Shimizu et al., 2006; Hoyer et al., 2008b; Hyvärinen et al., 2010; Rolland et al., 2022), studies on real-world datasets such as from atmospheric science (Huang et al., 2021) and healthcare (Tu et al., 2019) show that inferring the causal graph from data remains a challenging problem in practice (Reisach et al., 2021). Hence, causal inference studies often rely on human experts to provide the causal graph.

In this paper, based on the fact that the *topological/causal order* over the graph variables is enough for effect inference, we leverage Large Language Models (LLMs) as virtual domain experts to propose an automated method to query causal order (and hence causal effect). Moreover, providing the order between variables is the right question to ask experts because order depends only on the variables under question, unlike existence of a graph edge that depends on which other variables are present (to account for direct and indirect effects). For example, consider the data-generating process, *lung cancer → doctor visit → positive Xray*. If an expert is asked whether there should be a causal edge from *lung cancer* to *positive Xray*, they would answer "Yes" (indeed, such an edge exists in the BNLearn *Cancer* dataset (Scutari & Denis, 2014)). However, if they are told that the set of observed variables additionally includes *doctor visit*, then the correct answer would be to not

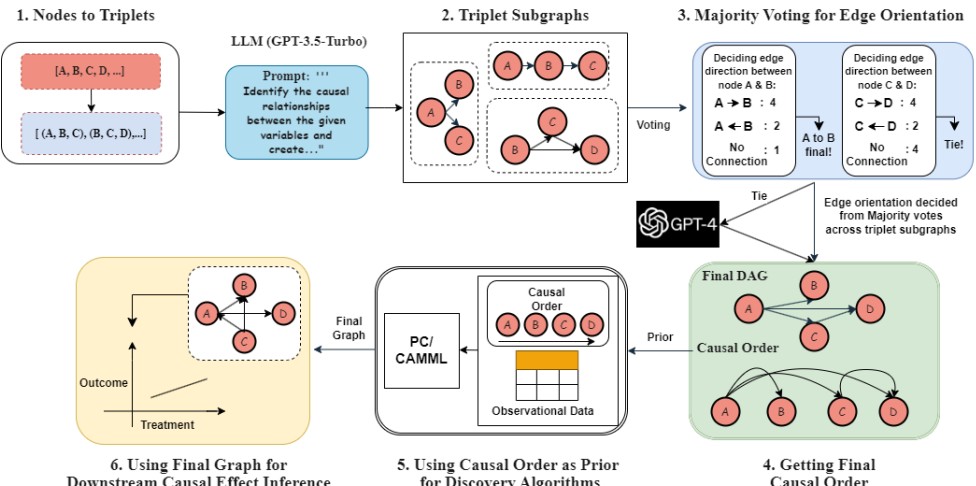

Figure 1: The *LLM-augmented* causal inference process based on inferring causal order. We propose a triplet-based prompting technique to infer all three-variable subgraphs and aggregate them using majority voting to produce a causal order. The causal order can then be used to identify a valid backdoor adjustment set. Ties in causal order are broken using another LLM (e.g., GPT-4). For robustness, LLM-generated causal order may be used in conjunction with discovery algorithms like PC or CaMML.

create a direct edge between lung cancer and positive Xray, but rather create edges mediated through *doctor visit*. However, note that the causal order, *lung cancer ≺ positive Xray* remains the same in both settings ($a ≺ b$ indicates that $a$ occurs before $b$ in a casual process).

We show that large language models (LLMs) like GPT-3.5 (Hagendorff et al., 2022) and GPT-4 (Peng et al., 2023) can be used to approximate experts' capability to provide causal order, thereby automating the process of causal inference. Extending results using LLMs for *pairwise* causal discovery (Kıcıman et al., 2023), we find that LLMs can provide accurate causal order for a diverse set of benchmark graphs. To do so, we propose a novel *triplet*-based prompting strategy that asks LLM to consider three variables at once, compared to the pairwise prompts employed in past work (Kıcıman et al., 2023; Willig et al., 2022; Long et al., 2023; 2022). Causal order outputted using the triplet-based prompt outperforms pairwise prompts across all benchmark datasets we considered. Importantly, it avoids cycles in the predicted order whereas LLM outputs from pairwise prompts often yield cycles.

Still, LLMs can exhibit unknown failure modes. Therefore, a more principled way is to adapt existing graph discovery algorithms to utilize LLM output. To this end, we present two algorithms based on constraint-based and score-based discovery algorithms respectively. The first uses causal order from an LLM to orient the undirected edges outputted by a constraint-based algorithm such as PC (Spirtes et al., 2000). The second algorithm utilizes the LLM causal order as a prior to a score-based algorithm like CaMML (Wallace et al., 1996). Results show that LLM-augmented algorithms outperform the base causal discovery algorithms in determining the causal order. The overall methodology is depicted in Figure 1. Our contributions can be summarized as follows.

- We argue that in causal effect estimation, querying a domain expert for a causal order is more principled than asking for exact causal structure among variables.
- We provide a novel prompting strategy based on triplets and show that LLMs like GPT-3.5 can be used to obtain causal order for a diverse range of datasets.
- We propose two algorithms combining causal discovery algorithms with LLM output and show that the final causal order is substantially more accurate than the discovery algorithms alone.

## 2 RELATED WORK

**Combining graph discovery and causal inference.** Historically, causal discovery and causal effect inference have been studied separately. Graph discovery algorithms can broadly be divided into (i) algorithms using conditional independence tests (*constraint-based*) (Glymour et al., 2019); (ii) algorithms using a score function to evaluate predicted graph (*score-based*) (Glymour et al., 2019); (iii)

algorithms that determine a causal order and then infer edges (*order-based*) (Rolland et al., 2022; Teyssier & Koller, 2005); and (iv) deep learning-based methods that formulate an optimization problem based on acyclicity and sparsity constraints (Zheng et al., 2018; Lachapelle et al., 2020). Causal discovery methods are evaluated on error with respect to the true graph, e.g., using the structural hamming distance (SHD) (Acid & de Campos, 2003; Tsamardinos et al., 2006). In contrast, causal inference methods focus on the estimation of causal effect given a causal graph (Pearl, 2009); the graphs are assumed to be known. A natural way to combine these approaches to use the graph outputted by discovery algorithms in inference methods, as in (Hoyer et al., 2008a; Mooij et al., 2016; Maathuis et al., 2010; Gupta et al., 2022). In this paper, we show that there exists a simpler way to combine the two approaches: only a causal order is needed instead of the full graph.

**Knowledge-driven Causal Discovery:** Prior knowledge has been used in causal discovery literature (Hasan & Gani, 2022; Constantinou et al., 2023; Heckerman & Geiger, 2013; Teshima & Sugiyama, 2021; O'Donnell et al., 2006; Wallace et al., 1996). These methods rely on prior knowledge such as domain expert opinions and documented knowledge from randomized controlled trials (RCT). Various priors have been studies in literature, including the priors of the form *edge existence*, *forbidden edge*, *ancestral constraints* (Constantinou et al., 2023; Ban et al., 2023). Prior knowledge significantly reduces the search space over all possible causal graphs.

Recent advancements in LLMs has led to more attention towards knowledge-driven causal discovery (Kıcıman et al., 2023; Ban et al., 2023; Long et al., 2023; Willig et al., 2022). Unlike causal discovery algorithms that use statistical patterns in the data, LLM-based algorithms use metadata such as variable names. Most of these methods use only LLMs to predict the causal relationships among a set of variables (Kıcıman et al., 2023; Willig et al., 2022; Long et al., 2022). Recent work also shows how LLMs can be used as priors or imperfect experts which can be combined with different types of discovery algorithms like (Long et al., 2023) uses LLMs to improve output of a constraint-based algorithm for full graph discovery by orienting undirected edges in the CPDAG and (Ban et al., 2023) uses LLMs as priors for scoring-based methods. However, the focus of these works has been on minimizing graph error metrics such as SHD. Instead, we focus on the downstream causal inference task and choose causal order as the metric since it directly correlates with accuracy in effect estimation whereas SHD does not.

**LLM Prompting Strategies for Causal Discovery:** Existing LLM-based algorithms for graph discovery (Kıcıman et al., 2023; Long et al., 2022; Ban et al., 2023) use a pairwise prompt, essentially asking "does A cause B" with varying levels of prompt complexity. Extending this line of work, we propose a triplet-based prompt that provides more accurate answers and avoids cycles when querying relationships between variables. As a result, our triplet-based prompt may be of independent interest to improve LLM-based graph discovery. We also explore the chain-of-thought prompting strategy (Wei et al., 2022) in our experiments.

# 3 BACKGROUND AND PROBLEM FORMULATION

Let $\mathcal{G}(\mathbf{X}, \mathbf{E})$ be a causal directed acyclic graph (DAG) consisting of a set of variables $\mathbf{X} = \{X_1, \ldots, X_n\}$ and a set of directed edges $\mathbf{E}$ among the variables in $\mathbf{X}$. A directed edge $X_i \rightarrow X_j \in \mathbf{E}$ denotes the *direct* causal influence of the variable $X_i$ on the variable $X_j$. Let $pa(X_i) = \{X_k | X_k \rightarrow X_i\}$, $de(X_i) = \{X_k | X_k \leftarrow \cdots \leftarrow X_i\}$ denote the set of *parents* and *descendants* of $X_i$ respectively. We focus on a downstream application of causal graph discovery called causal effect inference, defined as follows.

**Definition 3.1.** *(Average Causal Effect (Pearl, 2009)) The average causal effect (ACE) of a variable $X_i$ on a variable $X_j$ is defined as*

$$ACE_{X_i}^{X_j} = \mathbb{E}[X_j | do(X_i = x_i)] - \mathbb{E}[X_j | do(X_i = x_i^*)] \tag{1}$$

In Defn 3.1, $X_i$ is called the *treatment* variable and $X_j$ is called the *target* variable. $do(X_i = x_i)$ denotes an external intervention to the variable $X_i$ with the value $x_i$. The interventional quantity $\mathbb{E}[X_j | do(X_i = x_i)]$ is different from conditional $\mathbb{E}[X_j | X_i = x_i]$ since it involves setting the value of $X_i$ rather than conditioning on it. To estimate the quantity $\mathbb{E}[X_j | do(X_i = x_i)]$ from observational data, the backdoor adjustment formula is used.

**Definition 3.2.** *(Back-door Adjustment (Pearl, 2009)) Given a DAG $\mathcal{G}$, a set of variables $\mathbf{Z}$ satisfies back-door criterion relative to a pair of treatment and target variables $(X_i, X_j)$ if*

129    *(i) no variable in $\mathbf{Z}$ is a descendant of $X_i$; and*

130    *(ii) $\mathbf{Z}$ blocks every path between $X_i$ and $X_j$ that contains an arrow into $X_i$.*

131  where a *path* in a causal DAG is a sequence of unique vertices $X_i, X_{i+1}, \ldots, X_j$ with a directed
132  edge between each consecutive vertices $X_k$ and $X_{k+1}$ (either $X_k \rightarrow X_{k+1}$ or $X_{k+1} \rightarrow X_k$).
133  If a set of variables $\mathbf{Z}$ satisfies the back-door criterion relative to $(X_i, X_j)$, $\mathbb{E}[X_j|do(X_i = x_i)]$
134  can be computed using the formula: $\mathbb{E}[X_j|do(X_i = x_i)] = \mathbb{E}_{\mathbf{z} \sim \mathbf{Z}}\mathbb{E}[X_j|X_i = x_i, \mathbf{Z} = \mathbf{z}]$ (Thm.
135  3.3.2 of (Pearl, 2009)). To ensure that all variables in $\mathbf{Z}$ are observed, we assume that there are no
136  unobserved variables in the underlying causal graph.

## 4    CAUSAL ORDER IS SUFFICIENT FOR EFFECT ESTIMATION

138  Although backdoor adjustment is defined with respect to a DAG $\mathcal{G}$, we now show that the causal
139  order is sufficient to find a valid backdoor set. We also discuss why providing the causal order is a
140  better task for experts than providing the graph.

### 4.1    CAUSAL (TOPOLOGICAL) ORDER YIELDS A VALID BACKDOOR SET

142  **Definition 4.1.** *(**Topological Order.**) Given a causal graph $\mathcal{G}(\mathbf{X}, \mathbf{E})$, a sequence $\pi$ of variables $\mathbf{X}$*
143  *is said to be a topological order iff for each edge $X_i \rightarrow X_j \in \mathbf{E}$, $\pi_i < \pi_j$.*

144  **Proposition 4.2.** *(Pearl, 2009; Cinelli et al., 2022) Under the no confounding assumption, given an*
145  *pair of treatment and target variables $(X_i, X_j)$ in $\mathcal{G}$, $\mathbf{Z} = \{X_k|\pi_k < \pi_i\}$ is a valid adjustment set*
146  *relative to $(X_i, X_j)$ for any topological ordering $\pi$ of $\mathcal{G}$.*

147  Proof of all Propositions are in Appendix § A. Propn 4.2 states that all the variables that precede the
148  treatment variable in a topological order $\pi$ of $\mathcal{G}$ constitute a valid adjustment set. Note that the set $\mathbf{Z}$
149  may contain variables that are not necessary to adjust for, e.g., ancestors of only treatment or only
150  target variables. For statistical estimation, ancestors of target variable are beneficial for precision
151  whereas ancestors of treatment can be harmful (Cinelli et al., 2022). On balance though, causal
152  effect practitioners tend to include all confounders that do not violate the backdoor criterion; we are
153  following the same principle.

154  In practice, however, we may not know the true order. To evaluate the goodness of a given causal
155  order, we use the topological divergence metric from (Rolland et al., 2022) (for an example, see Fig-
156  ure 3). The topological divergence of an estimated topological order $\hat{\pi}$ with ground truth adjacency

157  matrix $A$, denoted by $D_{top}(\hat{\pi}, A)$, is defined as $D_{top}(\hat{\pi}, A) = \sum_{i=1}^{n} \sum_{j:\hat{\pi}_i > \hat{\pi}_j} A_{ij}$. Where $A_{ij} = 1$ if

158  there is a directed arrow from node $i$ to $j$ else $A_{ij} = 0$. $D_{top}(\hat{\pi}, A)$ counts the number of edges that
159  cannot be recovered due to estimated topological order $\hat{\pi}$.

### 4.2    TOPOLOGICAL DIVERGENCE IS THE CORRECT METRIC FOR EFFECT ESTIMATION

161  Below we show that $D_{top}$ is a valid metric to optimize for effect estimation: $D_{top} = 0$ for a topo-
162  logical order is equivalent to obtaining the correct backdoor adjustment set using Proposition 4.2.

163  **Proposition 4.3.** *For an estimated topological order $\hat{\pi}$ and a true topological order $\pi$ of a causal*
164  *DAG $\mathcal{G}$ with the corresponding adjacency matrix $A$, $D_{top}(\hat{\pi}, A) = 0$ iff $\mathbf{Z} = \{X_k|\hat{\pi}_k < \hat{\pi}_i\}$ is a*
165  *valid adjustment set relative to $(X_i, X_j)$, $\forall \pi_i < \pi_j$.*

166  We now compare $D_{top}$ to structural hamming distance (SHD), a common metric used to evaluate
167  graph discovery algorithms. Given a true causal DAG $\mathcal{G}$ and an estimated causal DAG $\hat{\mathcal{G}}$, SHD
168  counts the number of missing, falsely detected, and falsely directed edges in $\hat{\mathcal{G}}$. Formally, $D_{top}$ acts
169  as a lower-bound on the structural hamming distance (SHD) (Rolland et al., 2022). However, as
170  we show below, SHD is not a good metric for evaluating downstream effect estimation accuracy.
171  Specifically, we show that SHD can be very high even when $D_{top} = 0$ and a valid backdoor set
172  can be inferred. This result is of significance since most estimated graphs (included those that are
173  LLM-generated (Ban et al., 2023; Long et al., 2023)) are evaluated on SHD.

174  **Definition 4.4.** *(**Level Order.**) Given a causal graph $\mathcal{G}(\mathbf{X}, \mathbf{E})$, the level order refers to a systematic*
175  *assignment of levels to variables. This assignment begins with the set of variables $\{X_i|pa(X_i) = \emptyset\}$*
176  *at level 0. Subsequently, each of the remaining variables is assigned a level $i$ such that all nodes*
177  *within a given level $i$ has a directed path of length $i$ from one/more nodes in level 0.*

**Proposition 4.5.** *In a causal DAG $\mathcal{G}$ with $N$ levels in the level-ordering of variables where the level $i$ contains $n_i$ variables, $\exists \hat{\mathcal{G}}$ s.t. $SHD(\hat{\mathcal{G}}, \mathcal{G}) \geq \sum_{i=1}^{N-1} (n_i \times \sum_{j=i+1}^{N} n_j) - |\mathbf{E}|$ and $D_{top}(\hat{\pi}, A) = 0 \, \forall \hat{\pi}$ of $\hat{\mathcal{G}}$.*

Figure 2 shows the unsuitability of SHD for our work empirically. Given a fixed number of nodes, we sample a graph at random as the "ground-truth" and then consider all graph orientations of the same size (number of nodes) such that $D_{top} = 0$ with respect to to ground-truth graph. For these set of graphs, we compute the SHD with respect to the ground-truth graph. Notice that SHD exhibits high variance. For graphs with six nodes, SHD can vary from 0 to 14 even as $D_{top} = 0$ and backdoor set validity stays the same. Figure 3 shows this phenomenon on a real-world BNLearn dataset, *Cancer*. The candidate graph (right panel) has $D_{top} = 0$ with respect to the true graph (left)

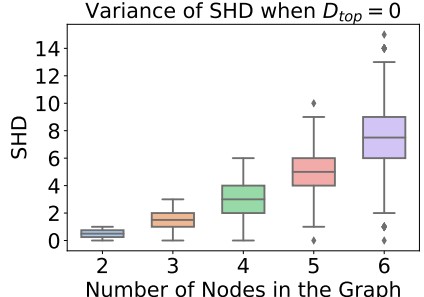

Figure 2: Variability of SHD for various graph sizes with consistent $D_{top} = 0$ within each graph.

and yields valid backdoor identification sets. However, its SHD is high (6), showing the disconnect between SHD and causal effect identification.

### 4.3 CAUSAL ORDER IS MORE SUITABLE TO ELICIT FROM EXPERTS THAN GRAPH EDGES

In addition to the favorable theoretical properties, causal order is easier to elicit from experts and can be objectively evaluated. This is because given two variables, their relative causal order does not depend on other variables whereas existence of an edge between them depends on which other variables are considered. To see this, let us continue the example from Figure 3 (left) where *pollution* causes *dyspnoea* (breathing difficulty) through the intermediary node *cancer*. Whether an edge exists between pollution and dyspnoea depends on whether *Cancer* variable is part of the study. In case an expert is only provided *pollution* and dyspnoea, they may add an edge between the

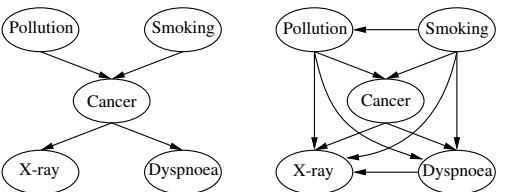

Figure 3: **Left:** Causal graph of Cancer dataset. **Right:** GPT-3.5's estimated causal graph of Cancer dataset. GPT-3.5 gets causal order correct at the cost of higher SHD score, which is not a relevant metric for causal inference. Here $D_{top} = 0$ and $SHD = 6$, showing the importance of $D_{top}$ in our study as compared to SHD.

two, but if *Cancer* node is also provided, they may not. Apriori, it is difficult to know which nodes may be relevant for a pair; hence experts' answers may not be consistent for questions about edges, but will always be consistent on causal order.

## 5 LARGE LANGUAGE MODELS AS VIRTUAL EXPERTS FOR CAUSAL ORDER

While causal order is a simpler construct than the graph, it still needs to be provided by a domain expert. We now study whether LLMs can used to obtain causal order, thereby making automating the process. We propose two kinds of prompting strategies; 1) Pairwise; and 2) Triplet-based. These methods employ variable names or extra metadata without utilizing the associated data.

### 5.1 PROMPT TECHNIQUES BASED ON A PAIR OF VARIABLES

A natural way to elicit causal order from LLMs is to ask about each pair of variables and aggregate the result. A similar pairwise strategy has been adopted by past work on inferring graph edges (Kıcıman et al., 2023; Ban et al., 2023; Long et al., 2022). Here we augment such strategies with additional contextual information. Our hypothesis is that adding context relevant to the pair of variables may help increase the accuracy of the LLM answers.

We propose four types of pairwise prompts (see Appendix § D for full prompts).

- **Basic prompt.** This is the simplest technique. We directly ask LLM to find the causal direction between a given pair of variables (Kıcıman et al., 2023).

- **Iterative Context.** Here we provide the previously oriented pairs as context in the prompt. Since the LLM has access to its previous decisions, we expect that it may avoid creating cycles through its predictions.
- **Markov Blanket Context.** Providing previously oriented pairs may become prohibitive for large graphs. Using the fact that a variable is independent of all other nodes given the Markov Blanket (Pearl, 2009), here we provide the Markov Blanket of the given node pairs as additional context in the prompt.
- **Chain-of-Thought (+In-context learning).** Based on encouraging results of providing in-context examples in the prompt for various tasks (Brown et al., 2020), here we include 3 examples of the ordering task that we expect the LLM to perform. Effectively, we provide example node pairs and their correct causal ordering before asking the question about the given nodes. Each example answer also contains an explanation of the answer, generated using Bing GPT-4. Adding the explanation encourages LLM to employ chain-of-thought reasoning (Wei et al., 2022) when deciding the causal order. To avoid overfitting, we select node pairs from graphs that are not evaluated in our study. Node pairs with and without direct edges were equally chosen. Examples of LLM's answers (and their explanations) using the CoT prompt can be found in Table A13 and Table A14 in Appendix.

### 5.2 PROMPT TECHNIQUE BASED ON TRIPLETS

As we shall see, while pairwise prompts are conceptually simple, they are prone to yielding cycles in the graph since they decide about each edge separately. Taking inspiration from the PC algorithm that employs constraints over three variables , we now describe a prompting technique based on iterating over all possible triplets given a set of nodes. Once the LLM has provided subgraphs for each triplet, we determine causal order between a pair by aggregating over all triplet LLM answers where the pair was included. To resolve ties, we use another LLM. The algorithm is as follows:

- From a given set of nodes in the graph, we generate all possible triplets, each triplet treated as independently from the others.
- We prompt the LLM to orient nodes of each triplet group to form a Directed Acyclic Graph representing the causal relationship between the nodes of the triplet. This will result in multiple mini graphs representing causal relationship for each triplet group.
- LLMs will be prompted to directly orient the three nodes for each triplet, hence identifying causal relationship based on the third node, similar to how PC functions.
- Once we have DAGs representing each triplet, we focus on merging them. Our Merging process can be broken down into two major steps:
    - We iterate over all node pairs, and for each combination we check what is the majority orientation between them over all the triplets containing the node pair.
    - In case there occurs a clash (same number of votes) between any of the two or all three possible edge orientation (A → B; B → A; No connection between A and B nodes), we resort to GPT-4 to resolve the clash by taking the final decision via CoT prompt.

In addition, the triplet prompt uses the techniques of in-context examples and chain-of-thought from the pairwise setup. An example prompt is shown in Table A12.

## 6 ALGORITHMS FOR LLM-GUIDED CAUSAL DISCOVERY

LLMs using the above prompts may yield accurate causal order predictions, but may also exhibit some unknown failure modes (Kıcıman et al., 2023). To increase robustness of the final result, we now provide algorithms for combining LLM-outputted causal order with existing causal discovery paradigms: (i) constraint-based and (ii) score-based methods.

### 6.1 CONSTRAINT-BASED METHODS USING POST-HOC EDGE ORIENTATION BY LLMS

Constraint-based algorithms return a graph where some edges may not be oriented. Given a graph from constraint-based algorithm like PC, we use the causal order $\hat{\pi}$ from LLM to orient the undirected edges. Iterating over the undirected edges, we first check if the nodes of that edge are occurring in $\hat{\pi}$. If yes, we orient the edge according to the causal order. Since there is a possibility that LLM's final graph might have some isolated nodes which won't be in $\hat{\pi}$, therefore if either (or both) nodes of the undirected edge are not included in $\hat{\pi}$, we query GPT-4 using pairwise CoT prompt (from Section 5.1) to finalise a direction between the pair.

---

**Algorithm 1** Combining constraint based methods and experts to get $\hat{\pi}$ for a given set of variables.

---

1: **Input:** LLM topological ordering $\hat{\pi}$, Expert $\mathcal{E}_{GPT4}$, PC-CPDAG $\hat{\mathcal{G}}$
2: **Output:** Estimated topological order $\hat{\pi}_{\text{final}}$ of $\{X_1, \ldots, X_n\}$.
3: **for** $(i - j) \in$ undirected-edges($\hat{\mathcal{G}}$) **do**
4:     If both the node $i$ and $j$ are in $\hat{\pi}$ and if $\hat{\pi}_i < \hat{\pi}_j$, orient $(i - j)$ as $(i \rightarrow j)$ in $\hat{\mathcal{G}}$.
5:     Otherwise, use the expert $\mathcal{E}_{GPT4}$ with CoT prompt to orient the edge $(i - j)$.
6: **end for**
7: $\hat{\pi}_{\text{final}} =$ topological ordering of $\hat{\mathcal{G}}$
8: return $\hat{\pi}$

---

## 6.2 SCORE-BASED METHODS USING EXPERT PRIORS

We utilize the output of LLM as a prior in the score-based algorithms. We provide the level order of the causal graph returned by LLM as a prior for a score-based algorithm. Unlike a similar LLM-prior approach by Ban et al. (2023), where they combine the output of LLM and a score based method using an ancestral constraint as a prior, ours is a sequential approach where a score based algorithm starts with the order based constraint, aligning with our goal of recovering causal order among variables. Optionally, we can provide prior probability to control the influence of prior on the algorithm. Algorithm on the right outlines the steps to combine score based method and expert knowledge in terms of variables' level order.

---

**Algorithm 2** Combining score based methods and experts to get $\hat{\pi}$ for a given set of variables.

---

1: **Input:** $\mathcal{D}$, variables $\{X_1, \ldots, X_n\}$, Expert $\mathcal{E}$, Score based method $\mathcal{S}$, *Prior* probability $p$.
2: **Output:** Estimated topological order $\hat{\pi}$ of $\{X_1, \ldots, X_n\}$.
3: $\hat{\mathcal{G}} = \mathcal{E}(X_1, \ldots, X_n)$
4: $L =$ level order of $\hat{\mathcal{G}}$.
5: **for** cycle $C \in \hat{\mathcal{G}}$ **do**
6:     **for** node $\in C$ **do**
7:         $L(\text{node}) = \min(\text{level(c)} \; \forall c \in C)$
8:     **end for**
9: **end for**
10: $\hat{\mathcal{G}} = \mathcal{S}(\mathcal{D}, L, p)$
11: $\hat{\pi} =$ topological ordering of $\hat{\mathcal{G}}$
12: return $\hat{\pi}$

---

# 7 EXPERIMENTS AND RESULTS

To evaluate the accuracy of LLM-based algorithms on inferring causal order, we perform experiments on the benchmark datasets from Bayesian network repository (Scutari & Denis, 2014): Earthquake, Cancer, Survey, Asia, Asia modified (Asia-M), and Child. Asia-M is derived from Asia by removing the node *either* since it is not a node with a semantic meaning (see Appendix§ C for details). We also used a medium sized subset graph (refer Figure A6 in Appendix) from the Neuropathic dataset (Tu et al., 2019) used for pain diagnosis. Except Child (with 20 nodes) and Neuropathic subgraph (with 22 nodes), all other graphs are small-scale graphs with <10 nodes.

## 7.1 $D_{top}$ CORRELATES WITH EFFECT ESTIMATION ERROR

Before comparing methods on the $D_{top}$ metric, we first show that $D_{top}$ has a strong correlation with effect estimation error and hence is the correct metric for effect inference. Specifically, we study how the error in causal effect, $\epsilon_{ACE}$, changes as values of the metrics $SHD, D_{top}$ change. For the datasets Cancer, Asia and Survey, we consider $dyspnoea$, $dyspnoea$, and $Travel$ respectively as the target variables. In each graph, we evaluate causal effects of each variable on the target variable. We iterate through estimated causal graphs with different values of SHD and $D_{top}$ and report the mean absolute difference between estimated and true causal effects. As Table 1 shows, when $D_{top}$ is zero, effect error $\epsilon_{ACE}$ is also zero. And as $D_{top}$ increases (right panel), effect error increases. In contrast, SHD has no correlation with the $\epsilon_{ACE}$.

| Cancer | | | |
|---|---|---|---|
| $SHD$ vs. $\epsilon_{ACE} \mid D_{top} = 0$ | | $D_{top}$ vs. $\epsilon_{ACE} \mid SHD = 2$ | |
| $SHD$ | $\epsilon_{ACE}$ | $D_{top}$ | $\epsilon_{ACE}$ |
| 0 | 0.00 | 0 | 0.00 |
| 2 | 0.00 | 1 | 0.25 |
| 4 | 0.00 | 2 | 0.50 |
| **Asia** | | | |
| $SHD$ vs. $\epsilon_{ACE} \mid D_{top} = 0$ | | $D_{top}$ vs. $\epsilon_{ACE} \mid SHD = 3$ | |
| $SHD$ | $\epsilon_{ACE}$ | $D_{top}$ | $\epsilon_{ACE}$ |
| 0 | 0.00 | 1 | 0.14 |
| 6 | 0.00 | 2 | 0.22 |
| 10 | 0.00 | 3 | 0.57 |
| **Survey** | | | |
| $SHD$ vs. $\epsilon_{ACE} \mid D_{top} = 0$ | | $D_{top}$ vs. $\epsilon_{ACE} \mid SHD = 2$ | |
| $SHD$ | $\epsilon_{ACE}$ | $D_{top}$ | $\epsilon_{ACE}$ |
| 0 | 0.00 | 0 | 0.00 |
| 2 | 0.00 | 1 | 0.25 |
| 4 | 0.03 | 2 | 0.50 |

Table 1: $\epsilon_{ACE}$ vs. $SHD$ ($D_{top}$) given $D_{top}$ ($SHD$)

| Dataset | $D_{top}$ | SHD | IN/TN | Cycles |
|---------|-----------|-----|-------|--------|
| Base Prompt | | | | |
| Earthquake | 0 | 7 | 0/5 | 0 |
| Cancer | 0 | 6 | 0/5 | 0 |
| Survey | 3 | 12 | 0/6 | 0 |
| Asia | - | 21 | 0/8 | 7 |
| Asia-M | - | 15 | 0/7 | 6 |
| Child | - | 177 | 0/20 | 20 |
| Neuropathic | - | 212 | 0/22 | 22 |
| All Directed Edges | | | | |
| Earthquake | 1 | 9 | 0/5 | 0 |
| Cancer | 1 | 7 | 0/5 | 0 |
| Survey | 2 | 11 | 0/6 | 0 |
| Asia | - | 21 | 0/8 | 8 |
| Asia-M | 0 | 13 | 0/7 | 0 |
| Child | - | 139 | 0/20 | 18 |
| Neuropathic | - | 194 | 0/22 | 20 |
| Markov Blanket | | | | |
| Earthquake | 0 | 8 | 0/5 | 0 |
| Cancer | 0 | 6 | 0/5 | 0 |
| Survey | 3 | 12 | 0/6 | 0 |
| Asia | - | 21 | 0/8 | 5 |
| Asia-M | 0 | 14 | 0/7 | 0 |
| Child | - | 167 | 0/20 | 20 |
| Neuropathic | - | 204 | 0/22 | 21 |

Table 2: Comparison of various prompting strategies for only LLM based setups, providing different contextual cues in each setup about the graph. IN: Isolated Nodes, TN:Total Nodes.

| Dataset | $D_{top}$ | SHD | IN/TN | Cycles |
|---------|-----------|-----|-------|--------|
| Chain of Thought | | | | |
| Earthquake | 1 | 4 | 0/5 | 0 |
| Survey | 1 | 6 | 2/6 | 0 |
| Asia | 1 | 17 | 0/8 | 0 |
| Asia-M | 1 | 11 | 0/7 | 0 |
| Child | - | 91 | 0/20 | 13 |
| Neuropathic | - | 64 | 0/22 | 8 |
| Triplet Prompt | | | | |
| Earthquake | 0 | 4 | 0/5 | 0 |
| Cancer | 0 | 4 | 1/5 | 0 |
| Survey | 0 | 6 | 0/6 | 0 |
| Asia | 1 | 7 | 1/8 | 0 |
| Asia-M | 0 | 3 | 2/7 | 0 |
| Child | 0 | 29 | 11/20 | 0 |
| Neuropathic | 2 | 23 | 16/22 | 0 |

Table 3: Triplet Prompt output using variable names with their descriptions (Cancer not included since CoT prompt has examples from this graph). IN: Isolated Nodes, TN:Total Nodes.

## 7.2 TRIPLET PROMPTING TECHNIQUE IS MOST ACCURATE FOR CAUSAL ORDER

Tables 2 and 3 compare the different prompting techniques. As the graph size increases, we observe limitations with pairwise prompts. In many cases, pairwise prompts yield cycles in many cases due to which $D_{top}$ cannot be computed. In particular, for Child dataset with 20 nodes, pairwise prompts yield anywhere from 13-79 cycles. LLM output tends to connect more edges than needed, which explains why SHD is high. Overall, among the pairwise prompts, the chain of thought prompt performs the best: it has the lowest $D_{top}$ on the four small graphs and the lowest number of cycles for Child and Neuropathic datasets. This indicates that in-context examples and chain-of-thought reasoning helps to increase accuracy of causal order output, but other contextual cues do not matter.

Finally, the triplet prompt provides the most accurate causal order. Even for medium-size graphs like Child and Neuropathic, the LLM output includes no cycles and SHD is fairly low betwen 4-29. Moreover, $D_{top}$ is zero for all datasets, except for Asia and Neuropathic where it is 1 and 2 respectively. That said, we do see that isolated nodes in the output increase compared to the pairwise prompts (all graphs are connected, so outputting an isolated node is an error). Considering LLMs as virtual experts, this indicates that there are some nodes on which the LLM expert cannot determine the causal order. This is still a better tradeoff than outputting the wrong causal order, which can confuse downstream algorithms. Overall, therefore, we conclude that the triplet prompt provides the most robust causal order predictions.

## 7.3 LLMS IMPROVE CAUSAL ORDER ACCURACY OF EXISTING DISCOVERY ALGORITHMS

We now study whether LLM output can be used to increase accuracy of discovery algorithms in inferring causal order. We compare with popular causal discovery methods: PC (Spirtes et al., 2000), SCORE (Rolland et al., 2022), ICA-LiNGAM (Shimizu et al., 2006), Direct-LiNGAM (Shimizu et al., 2011), NOTEARS (Zheng et al., 2018), and Causal discovery via minimum message length (CaMML) (Wallace et al., 1996); across five different sample sizes: 250, 500, 1000, 5000, 10000. For LLM, we use the triplet prompt. Table 4 shows the $D_{top}$ metric for different algorithms and compares it to the $D_{top}$ of our combined algorithms: PC+LLM and CaMML+LLM. Among the discovery algorithms, we find that PC and CaMML perform the best, with the lowest $D_{top}$ across the five datasets. For Neuropathic dataset, ICA LiNGAM is also competitive.

| N | Dataset | PC | SCORE | ICA LiNGAM | Direct LiNGAM | NOTEARS | CaMML | Ours (PC+LLM) | Ours (CaMML+LLM) |
|---|---|---|---|---|---|---|---|---|---|
| N = 250 | Earthquake | 0.30±0.44 | 4.00±0.00 | 3.20±0.39 | 3.00±0.00 | 1.80±0.74 | 2.00±0.00 | **0.00±0.00** | **0.00±0.00** |
| | Cancer | **0.00±0.00** | 3.00±0.00 | 4.00±0.00 | 3.60±0.48 | 2.00±0.00 | 2.00±0.00 | **0.00±0.00** | **0.00±0.00** |
| | Survey | 0.50±0.00 | 3.00±0.00 | 6.00±0.00 | 6.00±0.00 | 3.20±0.39 | 3.33±0.94 | **0.00±0.00** | 3.33±0.94 |
| | Asia | 2.33±0.59 | 5.00±0.00 | 6.20±0.74 | 7.00±0.00 | 4.00±0.00 | 1.85±0.58 | 0.00±0.00 | **0.97±0.62** |
| | Asia-M | 2.00±0.00 | 5.00±0.00 | 7.60±0.48 | 6.20±1.16 | 3.40±0.48 | 1.00±0.00 | **0.00±0.00** | 1.71±0.45 |
| | Child | 8.16±1.58 | 8.80±2.70 | 12.8±0.97 | 13.0±0.63 | 15.0±1.09 | **3.00±0.00** | 4.00±0.00 | 3.53±0.45 |
| | Neuropathic | 3.25±0.00 | 6.00±0.00 | 13.0±6.16 | 10.0±0.00 | 9.00±0.00 | 10.4±1.95 | **1.00±0.00** | 5.00±0.00 |
| N = 500 | Earthquake | 0.85±0.65 | 4.00±0.00 | 3.20±0.39 | 3.40±0.48 | 1.20±0.40 | **0.00±0.00** | 0.4±0.89 | **0.00±0.00** |
| | Cancer | **0.00±0.00** | 3.00±0.00 | 3.40±0.48 | 3.00±0.00 | 2.00±0.00 | 1.00±0.00 | **0.00±0.00** | 1.00±0.00 |
| | Survey | 1.75±0.00 | 4.00±0.00 | 6.00±0.0 | 6.00±0.00 | 3.40±0.48 | 3.39±0.08 | **1.00±0.00** | 3.33±0.94 |
| | Asia | 3.00±0.00 | 5.00±0.00 | 5.60±0.48 | 7.00±0.00 | 3.20±0.39 | 3.81±0.39 | 1.00±0.00 | **0.97±0.62** |
| | Asia-M | 2.00±0.00 | 6.00±0.00 | 7.60±0.48 | 5.00±0.00 | 3.80±0.39 | 2.00±0.00 | 1.00±0.00 | **0.17±0.45** |
| | Child | 9.79±1.17 | 6.20±1.32 | 12.2±0.74 | 10.6±1.35 | 15.4±0.48 | **2.00±0.00** | 4.6±1.34 | **2.00±0.00** |
| | Neuropathic | 7.50±0.00 | 6.00±0.00 | 9.00±1.41 | 13.0±0.00 | 11.0±0.00 | **5.32±0.57** | 8.00±0.00 | 7.49±0.64 |
| N = 1000 | Earthquake | 0.80±0.57 | 4.00±0.00 | 3.00±0.00 | 3.00±0.00 | 1.00±0.00 | **0.00±0.00** | 0.20±0.44 | **0.00±0.00** |
| | Cancer | **0.00±0.00** | 3.00±0.00 | 3.00±0.00 | 3.00±0.00 | 2.00±0.00 | 1.60±0.48 | **0.00±0.00** | **0.00±0.00** |
| | Survey | 1.00±0.00 | 4.00±0.00 | 5.80±0.39 | 5.40±0.48 | 3.20±0.39 | 2.71±0.27 | **1.00±0.00** | 2.83±0.00 |
| | Asia | 3.09±1.05 | 4.00±0.00 | 6.20±0.74 | 6.60±0.48 | 3.40±0.48 | 1.75±0.43 | 1.75±0.95 | **0.97±0.62** |
| | Asia-M | 2.50±0.00 | 4.00±0.00 | 8.00±0.00 | 5.20±0.39 | 3.40±0.48 | 2.04±0.51 | 2.00±0.00 | **0.65±0.47** |
| | Child | 9.61±1.07 | 3.80±0.74 | 12.2±1.72 | 11.8±0.74 | 15.2±0.97 | **2.00±0.00** | 8.0±0.00 | **2.00±0.40** |
| | Neuropathic | - | 6.00±0.00 | **4.00±0.81** | 12.0±0.00 | 12.0±0.00 | 5.54±0.75 | - | 10.1±2.12 |
| N = 5000 | Earthquake | 0.30±0.67 | 4.00±0.00 | 2.80±0.39 | 3.00±0.00 | 1.00±0.00 | 0.80±0.97 | **0.00±0.00** | **0.00±0.00** |
| | Cancer | **0.00±0.00** | 3.00±0.00 | 3.00±0.00 | 3.00±0.00 | 2.00±0.00 | 2.00±0.00 | **0.00±0.00** | **0.00±0.00** |
| | Survey | 2.00±0.00 | 4.00±0.00 | 5.00±0.00 | 5.00±0.00 | 3.00±0.00 | 3.33±0.69 | **2.00±0.00** | 2.60±0.00 |
| | Asia | 3.05±0.94 | 4.00±0.00 | 6.60±0.74 | 4.40±1.35 | 3.40±0.48 | 1.75±0.43 | 2.80±1.30 | **0.97±0.62** |
| | Asia-M | 1.00±0.00 | 4.00±0.00 | 7.60±0.48 | 4.60±0.48 | 3.20±0.39 | 1.68±0.46 | 0.20±0.44 | **0.00±0.00** |
| | Child | 8.42±0.75 | 3.00±0.00 | 12.6±0.79 | 10.8±1.72 | 14.2±0.40 | **3.00±0.00** | 7.00±0.00 | **3.00±0.00** |
| | Neuropathic | 9.00±0.00 | 6.00±0.00 | 9.33±0.94 | 10.0±0.00 | 10.0±0.00 | 4.20±0.96 | 9.00±0.00 | **1.23±0.42** |
| N = 10000 | Earthquake | **0.00±0.00** | 4.00±0.00 | 3.00±0.00 | 3.00±0.00 | 1.00±0.00 | 0.40±0.48 | **0.00±0.00** | **0.00±0.00** |
| | Cancer | **0.00±0.00** | 3.00±0.00 | 3.00±0.00 | 3.00±0.00 | 2.00±0.00 | 0.60±0.80 | **0.00±0.00** | **0.00±0.00** |
| | Survey | 2.00±0.00 | 4.00±0.00 | 5.00±0.00 | 5.00±0.00 | 3.00±0.00 | 3.60±1.35 | 2.00±0.00 | **1.83±0.00** |
| | Asia | 1.95±0.41 | 4.00±0.00 | 6.00±0.00 | 4.40±1.35 | 3.00±0.00 | 1.40±0.48 | 1.20±0.83 | **0.34±0.47** |
| | Asia-M | 1.75±0.00 | 4.00±0.00 | 8.00±0.00 | 4.80±0.39 | 3.00±0.00 | 2.00±0.00 | **0.00±0.00** | **0.00±0.00** |
| | Child | 7.67±0.65 | 3.00±0.00 | 12.2±1.46 | 11.6±0.48 | 14.4±0.48 | 2.80±0.84 | 7.00±0.00 | **1.00±0.00** |
| | Neuropathic | 10.00±0.00 | 6.00±0.00 | **1.00±0.00** | 10.0±0.00 | 10.0±0.00 | 3.00±0.00 | 10.00±0.00 | **1.00±0.00** |

Table 4: Comparison with existing discovery methods. Mean and std dev of $D_{top}$ over 5 runs. (For Neuropathic subgraph (1k samples), PC Algorithm returns cyclic graphs in the MEC)

For the BNLearn datasets, adding LLM output decreases the $D_{top}$ of both algorithms further. Specifically, PC+LLM leads to a significant reduction in $D_{top}$ and the gains are larger at lower sample sizes. This indicates that obtaining causal order from LLMs may matter more in limited sample settings. At sample size of 500, $D_{top}$ of PC is nearly double that of PC+LLM for most datasets. Going from CaMML to CaMML+LLM, we also see significant reductions in $D_{top}$. Interestingly, CaMML+LLM yields benefits even at higher sample sizes. At a sample size of 10,000, CaMML's $D_{top}$ for Child and Asia surpasses CaMML+LLM by three and fivefold respectively.

For the Neuropathic dataset, we see a similar pattern: adding LLM to existing algorithms improves $D_{top}$ or keeps it constant, except at sample sizes 500 and 1000 where CaMML+LLM yields a worse $D_{top}$ than CaMML alone. However, as sample size increases to 5000 and 10000, we do see that CaMML+LLM improves the $D_{top}$ substantially compared to CaMML. Overall, these results show that LLM output can signficantly improve the accuracy of existing causal discovery algorithms.

## 8 LIMITATIONS AND CONCLUSIONS

We presented causal order as a suitable metric for evaluating quality of causal graphs for downstream effect inference tasks. Using a novel formulation of LLM prompts based on triplets, we showed that LLMs can be useful in the generating accurate causal order, both individually and in combination with existing discovery algorithms. Our results point to techniques that can automate the causal inference process.

That said, our work has limitations. We studied LLMs utility on popular benchmarks which may have been partially memorized. It will be interesting to extend our experiments to diverse and bigger datasets. In addition, we studied only one downstream task (effect inference). Identifying the necessary graph metrics for tasks such as causal prediction and counterfactual inference will be useful future work.

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

APPENDIX

## A   PROOFS OF PROPOSITIONS

**Proposition 4.2.** *(Pearl, 2009; Cinelli et al., 2022) Under the no confounding assumption, given an pair of treatment and target variables $(X_i, X_j)$ in $\mathcal{G}$, $\mathbf{Z} = \{X_k | \pi_k < \pi_i\}$ is a valid adjustment set relative to $(X_i, X_j)$ for any topological ordering $\pi$ of $\mathcal{G}$.*

*Proof.* We need to show that the set $\mathbf{Z} = \{X_k | \pi_k < \pi_i\}$ satisfies the conditions (i) and (ii) in Defn 3.2. For any variable $X_k$ such that $\pi_k < \pi_i$, we have $X_k \notin de(X_i)$ and hence the condition (i) is satisfied. Additionally, for each $X_k \in pa(X_i)$ we have $\pi_k < \pi_i$ and hence $pa(X_i) \subseteq \mathbf{Z}$. Since $pa(X_i)$ blocks all paths from $X_i$ to $X_j$ that contains an arrow into $X_i$ (Peters & Bühlmann, 2015), $\mathbf{Z}$ satisfies condition (ii). $\square$

**Proposition 4.3.** *For an estimated topological order $\hat{\pi}$ and a true topological order $\pi$ of a causal DAG $\mathcal{G}$ with the corresponding adjacency matrix $A$, $D_{top}(\hat{\pi}, A) = 0$ iff $\mathbf{Z} = \{X_k | \hat{\pi}_k < \hat{\pi}_i\}$ is a valid adjustment set relative to $(X_i, X_j)$, $\forall \pi_i < \pi_j$.*

*Proof.* The statement of proposition is of the form $A \iff B$ with $A$ being "$D_{top}(\hat{\pi}, A) = 0$" and $B$ being "$\mathbf{Z} = \{X_k | \hat{\pi}_k < \hat{\pi}_i\}$ is a valid adjustment set relative to $(X_i, X_j)$, $\forall i, j$". We prove $A \iff B$ by proving (i) $A \implies B$ and (ii) $B \implies A$.

(i) Proof of $A \implies B$: If $D_{top}(\hat{\pi}, A) = 0$, for all pairs of nodes $(X_i, X_j)$, we have $\hat{\pi}_i < \hat{\pi}_j$ whenever $\pi_i < \pi_j$. That is, causal order in estimated graph is same that of the causal order in true graph. Hence, from Propn 4.2, $\mathbf{Z} = \{X_k | \hat{\pi}_k < \hat{\pi}_i\}$ is a valid adjustment set relative to $(X_i, X_j)$, $\forall i, j$.

(ii) Proof of $B \implies A$: we prove the logical equivalent form of $B \implies A$ i.e., $\neg A \implies \neg B$, the *contrapositive* of $B \implies A$. To this end, assume $D_{top}(\hat{\pi}, A) \neq 0$, then there will be at least one edge $X_i \to X_j$ that cannot be oriented correctly due to the estimated topological order $\hat{\pi}$. i.e., $\hat{\pi}_j < \hat{\pi}_i$ but $\pi_j > \pi_i$. Hence, to find the causal effect of $X_i$ on $X_l$; $l \neq j$, $X_j$ is included in the back-door adjustment set $\mathbf{Z}$ relative to $(X_i, X_l)$. Adding $X_j$ to $\mathbf{Z}$ renders $\mathbf{Z}$ an invalid adjustment set because it violates the condition (i) of Defn 3.2. $\square$

**Proposition 4.5.** *In a causal DAG $\mathcal{G}$ with $N$ levels in the level-ordering of variables where the level $i$ contains $n_i$ variables, $\exists \hat{\mathcal{G}}$ s.t. $SHD(\hat{\mathcal{G}}, \mathcal{G}) \geq \sum_{i=1}^{N-1} (n_i \times \sum_{j=i+1}^{N} n_j) - |\mathbf{E}|$ and $D_{top}(\hat{\pi}, A) = 0 \,\forall \hat{\pi}$ of $\hat{\mathcal{G}}$.*

*Proof.* Recall that SHD counts the number of missing, falsely detected, and falsely directed edges in the estimated causal graph as compared to the ground truth graph. Since we want $D_{top}(\hat{\pi}, A) = 0$; $\forall \hat{\pi}$ of $\hat{\mathcal{G}}$, there cannot be an edge $X_i \to X_j$ in $\hat{\mathcal{G}}$ such that $X_i \leftarrow X_j$ is in $\mathcal{G}$. This constraint avoids the possibility of having falsely directed edges in $\hat{\mathcal{G}}$. Consider a $\hat{\mathcal{G}}$ with all the edges in $\mathcal{G}$ and in addition, each variable in level $i$ having a directed edge to each variable in all levels below level $i$. All such edges contribute to the SHD score while still obeying the causal ordering in $\mathcal{G}$. This number will be equal to $\sum_{i=1}^{N-1} (n_i \times \sum_{j=i+1}^{N} n_j) - |\mathbf{E}|$. The quantity $\sum_{i=1}^{N-1} (n_i \times \sum_{j=i+1}^{N} n_j)$ is the number of edges possible from each node to the every other node in the levels below it. We need to subtract the number of existing edges in $\mathbf{E}$ to count the newly added edges that contribute to the SHD score. Now, we can remove some of the edges $X_i \to X_j$ from $\hat{\mathcal{G}}$ such that $X_i \to X_j$ is in $\mathcal{G}$ while still leading to same causal ordering of variables. This leads to increased SHD score due to missing edges in $\hat{\mathcal{G}}$. Since it will only increase the SHD score, we ignore such corner cases. $\square$

## B   ADDITIONAL RESULTS

Table A1 shows the results of various prompt strategies and their improvements over no-prior methods.

| | Dataset | PC | CaMML | Ours (CoT) (PC+LLM) | Ours (CoT) (CaMML+LLM) | Ours (Triplet Pairwise) (PC+LLM) | Ours (Triplet Pairwise) (CaMML+LLM) |
|---|---|---|---|---|---|---|---|
| $N=250$ | Earthquake | 0.30±0.44 | 2.00±0.00 | **0.00±0.00** | 2.75±0.43 | 0.2±0.44 | **0.00±0.00** |
| | Cancer | **0.00±0.00** | 2.00±0.00 | **0.00±0.00** | **0.00±0.00** | **0.00±0.00** | **0.00±0.00** |
| | Survey | 0.50±0.00 | 3.33±0.94 | **0.00±0.00** | 3.33±0.94 | **0.00±0.00** | 2.66±0.94 |
| | Asia | 2.33±0.59 | 1.85±0.58 | **0.00±0.00** | 0.97±0.62 | 1.00±0.00 | 3.39±0.62 |
| | Asia-M | 2.00±0.00 | 1.00±0.00 | **0.00±0.00** | 1.71±0.45 | **0.00±0.00** | 1.71±0.45 |
| | Child | 8.16± 1.58 | **3.00±0.00** | 4.70±1.34 | 3.53±0.45 | 8.00±3.17 | 4.00±0.00 |
| | Neuropathic | 3.25±0.00 | 10.4±1.95 | 1.00±0.00 | 8.00±0.00 | **0.00±0.00** | 12.63±1.92 |
| $N=500$ | Earthquake | 0.85±0.65 | **0.00±0.00** | 0.4±0.89 | 2.60±0.48 | 1.00±0.00 | **0.00±0.00** |
| | Cancer | **0.00±0.00** | 3.00±0.00 | **0.00±0.00** | 1.00±0.00 | **0.00±0.00** | **0.00±0.00** |
| | Survey | 1.75±0.00 | 3.39±0.08 | 1.00±0.00 | 3.33±0.94 | **0.60±0.00** | 2.66±0.94 |
| | Asia | 3.00±0.00 | 3.81±0.39 | 1.00±0.00 | 0.97±0.62 | **0.00±0.00** | 3.28±0.64 |
| | Asia-M | 2.00±0.00 | 2.00±0.00 | 1.00±0.00 | **0.17±0.45** | 1.00±0.00 | 1.04±0.20 |
| | Child | 9.79±1.17 | **2.00±0.00** | 4.60±1.34 | **2.00±0.00** | 9.20±2.16 | 3.00±0.00 |
| | Neuropathic | 7.50±0.00 | **5.32±0.57** | 9.00±0.00 | 8.90±0.00 | 9.00±0.00 | 12.1±1.56 |
| $N=1000$ | Earthquake | 0.80±0.57 | **0.00±0.00** | 0.20±0.44 | 2.00±0.00 | 1.40±1.31 | **0.00±0.00** |
| | Cancer | **0.00±0.00** | 2.00±0.00 | **0.00±0.00** | **0.00±0.00** | **0.00±0.00** | **0.00±0.00** |
| | Survey | 1.00±0.00 | 2.71±0.27 | **1.00±0.00** | 2.83±0.00 | 1.00±0.00 | 2.16±0.74 |
| | Asia | 3.09±1.05 | 1.75±0.43 | 1.75±0.95 | **0.97±0.62** | 3.00±0.00 | 1.78±0.41 |
| | Asia-M | 2.50±0.00 | 2.04±0.51 | 2.00±0.00 | **0.65±0.47** | 1.00±0.00 | **0.65±0.47** |
| | Child | 9.61±1.07 | **2.00±0.00** | 8.00±0.00 | **2.00±0.40** | 6.6±1.14 | 2.83±0.00 |
| | Neuropathic | - | 5.54±0.75 | - | 6.00±0.00 | - | **4.00±0.00** |
| $N=5000$ | Earthquake | 0.30±0.67 | 0.80±0.97 | **0.00±0.00** | 2.00±0.00 | **0.00±0.00** | **0.00±0.00** |
| | Cancer | **0.00±0.00** | 2.00±0.00 | **0.00±0.00** | **0.00±0.00** | **0.00±0.00** | **0.00±0.00** |
| | Survey | 2.00±0.00 | 3.33±0.69 | 2.00±0.00 | 2.60±0.00 | 2.00±0.00 | **1.80±0.83** |
| | Asia | 3.05±0.94 | 1.75±0.43 | 2.80±1.30 | 0.97±0.62 | **0.00±0.00** | 1.78±0.41 |
| | Asia-M | 1.00±0.00 | 1.68±0.46 | 0.20±0.44 | **0.00±0.00** | 2.00±0.00 | 1.00±0.00 |
| | Child | 8.42±0.75 | **3.00±0.00** | 7.00±0.00 | **3.00±0.00** | 7.4±1.51 | **1.00±0.00** |
| | Neuropathic | 9.00±0.00 | 4.20±0.96 | 9.00±0.00 | **3.00±0.00** | 9.00±0.00 | 3.31±0.00 |
| $N=10000$ | Earthquake | 0.00±0.00 | 0.40±0.48 | **0.00±0.00** | 2.00±0.00 | **0.00±0.00** | **0.00±0.00** |
| | Cancer | 0.00±0.00 | 2.00±0.00 | 0.60±0.80 | **0.00±0.00** | **0.00±0.00** | **0.00±0.00** |
| | Survey | 2.00±0.00 | 3.60±1.35 | 2.00±0.00 | 1.83±0.00 | 2.00±0.00 | **1.08±0.64** |
| | Asia | 1.95±0.41 | 1.40±0.48 | 1.20±0.83 | 0.34±0.47 | **0.00±0.00** | 1.78±0.41 |
| | Asia-M | 1.75±0.00 | 2.00±0.00 | **0.00±0.00** | **0.00±0.00** | **0.00±0.00** | 1.00±0.00 |
| | Child | 7.67±0.65 | 2.80±0.84 | 4.66±3.05 | **1.00±0.00** | 5.80±1.48 | **1.00±0.00** |
| | Neuropathic | 10.00±0.00 | 3.00±0.00 | - | 2.39±0.48 | 10.00±0.00 | **1.00±0.00** |

Table A1: $D_{top}$ metric results. Comparison with various prompting strategies. Neuropathic subgraph for 1k samples return cyclic graphs in the MEC. Using LLM CoT prior with Neuropathic 10k samples, orients the undirected edges to create cyclic graphs

### B.1 LLMs USED IN POST PROCESSING FOR GRAPH DISCOVERY

We conducted some experiments where we utilised discovery algorithms like PC for creating skeletons of the graph and employed LLMs for orienting the undirected edges. The idea was to utilise LLMs ability to correctly estimate the causal direction while leveraging PC algorithm's ability to give a skeleton which could be oriented in a post processing setup. We saw that LLM ended up giving improved results as compared to PC alone.

| Context | Base prompt | Past iteration orientations | Markov Blanket | PC (Avg. over MEC) |
|---|---|---|---|---|
| **1000 samples** | | | | |
| $D_{top}$ | 8.0 | 5.3 | 6.6 | 9.61 |
| SHD | 14.33 | 12.66 | 14.0 | 17.0 |
| **10000 samples** | | | | |
| $D_{top}$ | 6.33 | 9.66 | 6.0 | 7.67 |
| SHD | 9.0 | 13.33 | 8.33 | 12.0 |

Table A2: PC + LLM results where LLM is used to orient the undirected edges of the skeleton PC returns over different data sample sizes. We show how LLMs can be used in a post processing setup for edge orientation besides having the capability of acting as a strong prior for different discovery algorithms.

| Dataset | Number of Nodes | Number of Edges | Description (used as a context) |
|---|---|---|---|
| Asia | 8 | 8 | Model the possible respiratory problems someone can have who has recently visited Asia and is experiencing shortness of breath |
| Cancer | 5 | 4 | Model the relation between various variables responsible for causing Cancer and its possible outcomes |
| Earthquake | 5 | 5 | Model factors influencing the probability of a burglary |
| Survey | 6 | 6 | Model a hypothetical survey whose aim is to investigate the usage patterns of different means of transport |
| Child | 20 | 25 | Model congenital heart disease in babies |
| Neuropathic Pain Diagnosis (subgraph) | 22 | 25 | For neuropathic pain diagnosis |

Table A3: Overview of the datasets used.

## C    CAUSAL GRAPHS USED IN EXPERIMENTS

Figures A1-A5 show the causal graphs and details we considered from BNLearn repository (Scutari & Denis, 2014).

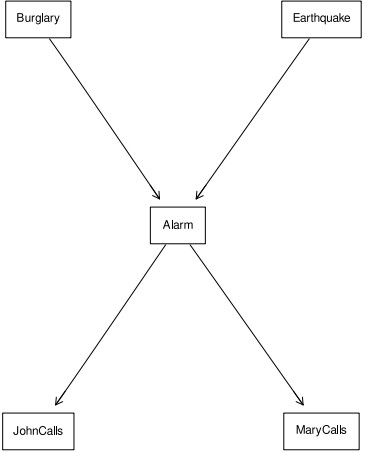

Figure A1: Earthquake Bayesian network. Abbreviations/Descriptions: Burglary: *burglar entering*, Earthquake: *earthquake hitting*, Alarm: *home alarm going off in a house*, JohnCalls: *first neighbor to call to inform the alarm sound*, Marycalls: *second neighbor to call to inform the alarm sound*.

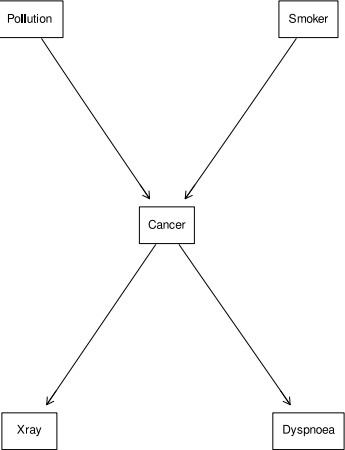

Figure A2: Cancer Bayesian network. Abbreviations/Descriptions: Pollution: *exposure to pollutants*, Smoker: *smoking habit*, Cancer: *Cancer*, Dyspnoea: *Dyspnoea*, Xray: *getting positive xray result*.

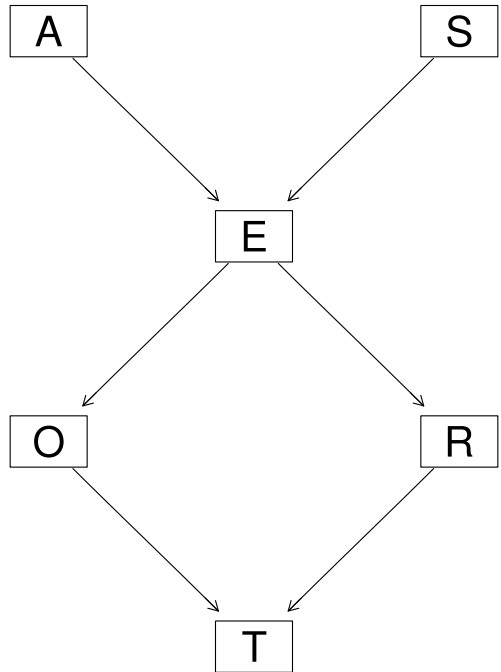

Figure A3: Survey Bayesian network. Abbreviations: A=*Age/Age of people using transport*, S=*Sex/male or female*, E=*Education/up to high school or university degree*, O=*Occupation/employee or self-employed*, R=*Residence/the size of the city the individual lives in, recorded as either small or big*, T=*Travel/the means of transport favoured by the individual*.

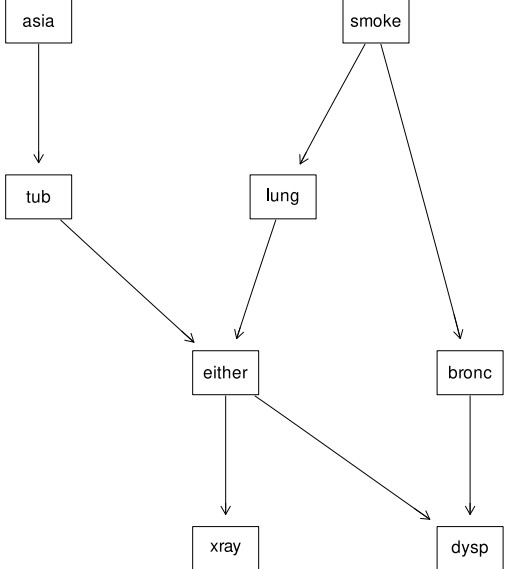

Figure A4: Asia Bayesian network. Abbreviations/Descriptions: asia=*visit to Asia/visiting Asian countries with high exposure to pollutants*, smoke=*smoking habit*, tub=*tuberculosis*, lung=*lung cancer*, either=*either tuberculosis or lung cancer*, bronc=*bronchitis*, dysp=*dyspnoea*, xray=*getting positve xray result*.

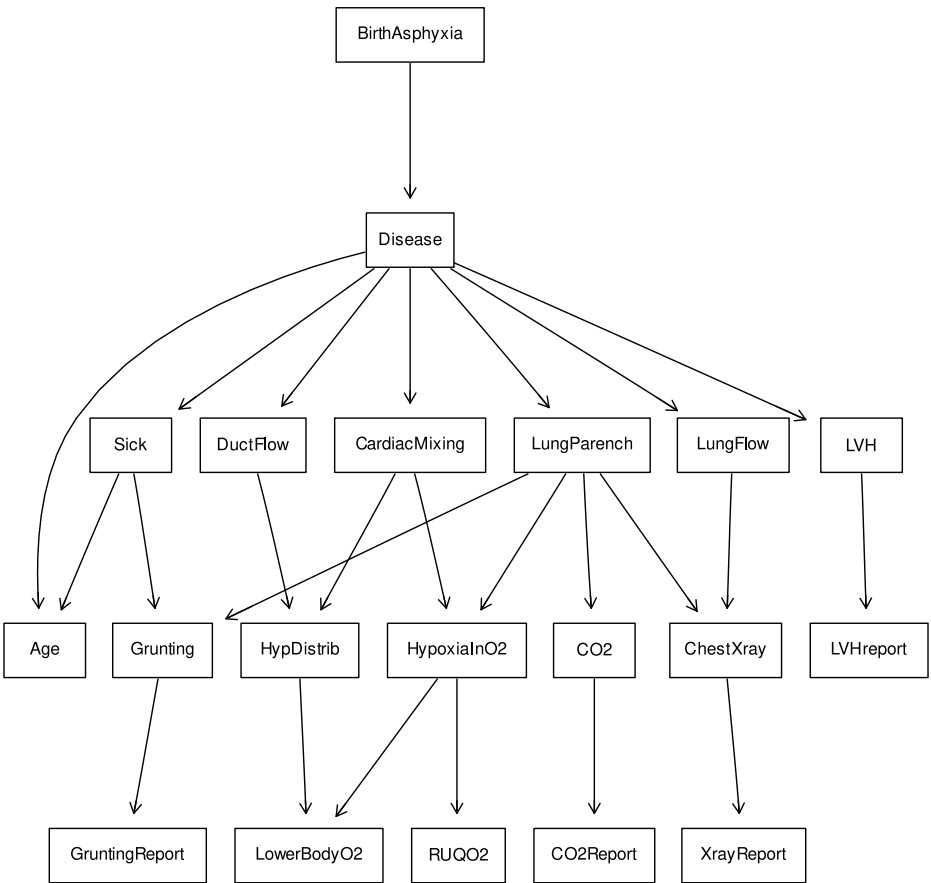

Figure A5: Child Bayesian network. Abbreviations: BirthAsphyxia: *Lack of oxygen to the blood during the infant's birth*, HypDistrib: *Low oxygen areas equally distributed around the body*, HypoxiaInO2: *Hypoxia when breathing oxygen*, CO2: *Level of carbon dioxide in the body*, ChestXray: *Having a chest x-ray*, Grunting: *Grunting in infants*, LVHreport: *Report of having left ventricular hypertrophy*, LowerBodyO2: *Level of oxygen in the lower body*, RUQO2: *Level of oxygen in the right upper quadricep muscle*, CO2Report: *A document reporting high levels of CO2 levels in blood*, XrayReport: *Report of having a chest x-ray*, Disease: *Presence of an illness*, GruntingReport: *Report of infant grunting*, Age: *Age of infant at disease presentation*, LVH: *Thickening of the left ventricle*, DuctFlow: *Blood flow across the ductus arteriosus*, CardiacMixing: *Mixing of oxygenated and deoxygenated blood*, LungParench: *The state of the blood vessels in the lungs*, LungFlow: *Low blood flow in the lungs*, Sick: *Presence of an illness*

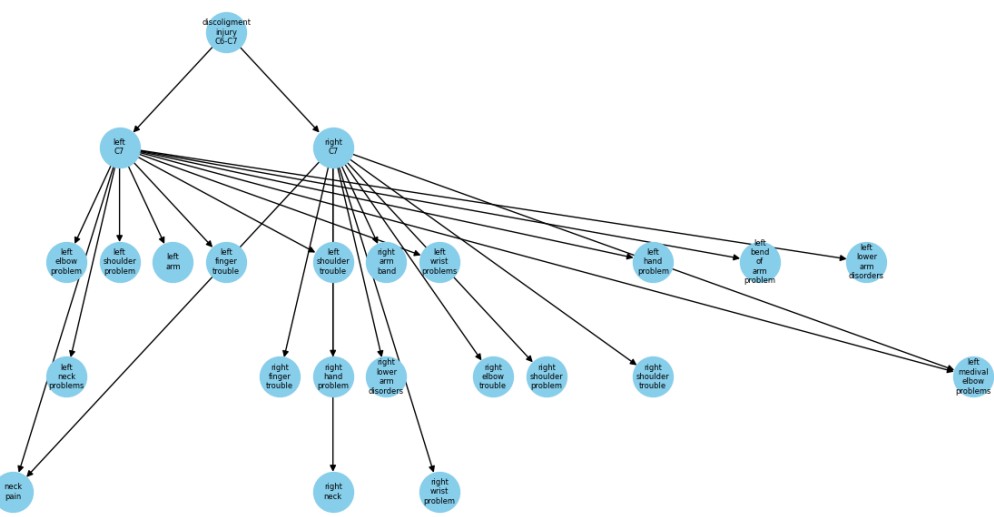

Figure A6: For Neuropathic dataset, we consider a sub-graph induced by one of the root nodes, containing the following 22 nodes and corresponding edges taken from https://observablehq.com/@turuibo/the-complete-causal-graph-of-neuropathic-pain-diagnosis: 'right C7', 'right elbow trouble', 'left shoulder trouble', 'left bend of arm problem', 'right shoulder trouble', 'right hand problem', 'left medival elbow problems', 'right finger trouble', 'left neck problems', 'left wrist problems', 'left shoulder problem', 'right neck', 'right wrist problem', 'right shoulder problem', 'discoligment injury C6 C7', 'left hand problem', 'left C7', 'right arm band', 'left lower arm disorders', 'neck pain', 'left finger trouble', 'left arm'. We did not use descriptions for the nodes of Neuropathic graph.

535 ## D  VARIOUS PROMPT STRATEGIES STUDIED

---

Question: For a causal graph used to model relationship of various factors and outcomes related to cancer with the following nodes: ['Pollution', 'Cancer', 'Smoker', 'Xray', 'Dyspnoea'],
Which cause-and-effect relationship is more likely between nodes 'smoker' and 'cancer'?

A. changing the state of node 'smoker' causally effects a change in another node 'cancer'.

B. changing the state of node 'cancer' causally effects a change in another node 'smoker'.

C. There is no causal relation between the nodes 'cancer' and 'smoker'.

Make sure to first provide a grounded reasoning for your answer and then provide the answer in the following format: <Answer>A/B/C</Answer>. It is very important that you output the final Causal graph within the tags like <Answer>A/B/C</Answer> otherwise your answer will not be processed.

Answer: The causal effect of "smoker" directing to "cancer" is based on the strong evidence from epidemiological studies linking smoking to an increased risk of developing cancer. Smoking introduces harmful substances into the respiratory system, leading to cellular damage and mutation, which significantly raises the likelihood of cancer development in the lungs or respiratory tract, subsequently impacting the occurrence of respiratory problems like shortness of breath. Therefore answer is <Answer>A</Answer>

Question: For a causal graph used to model relationship of various factors and outcomes related to cancer with the following nodes: ['Pollution', 'Cancer', 'Smoker', 'Xray', 'Dyspnoea'], Which cause-and-effect relationship is more likely between nodes 'xray' and 'dyspnoea'?

A. changing the state of node 'xray' causally effects a change in another node 'dyspnoea'.

B. changing the state of node 'dyspnoea' causally effects a change in another node 'xray'.

C. There is no causal relation between the nodes 'xray' and 'dyspnoea'.

Make sure to first provide a grounded reasoning for your answer and then provide the answer in the following format: <Answer>A/B/C</Answer>. It is very important that you output the final Causal graph within the tags like <Answer>A/B/C</Answer> otherwise your answer will not be processed.

Answer: Reasoning behind the lack of causal relation between X-ray and dyspnoea is that X-ray and dyspnoea are both effects of having cancer, but they do not directly cause or affect each other. X-ray is a diagnostic test that can help detect cancer in the lungs or other organs, while dyspnoea is a symptom of cancer that involves feeling short of breath. Therefore, X-ray and dyspnoea are not causally related, but they are both associated with cancer. Therefore answer is <Answer>C</Answer>

Question: For a causal graph used to model relationship of various factors and outcomes related to cancer with the following nodes: ['Pollution', 'Cancer', 'Smoker', 'Xray', 'Dyspnoea'], Which cause-and-effect relationship is more likely between nodes 'xray' and 'cancer'?

A. changing the state of node 'xray' causally effects a change in another node 'cancer'.

B. changing the state of node 'cancer' causally effects a change in another node 'xray'.

C. There is no causal relation between the nodes 'xray' and 'cancer'.

Make sure to first provide a grounded reasoning for your answer and then provide the answer in the following format: <Answer>A/B/C</Answer>. It is very important that you output the final Causal graph within the tags like <Answer>A/B/C</Answer> otherwise your answer will not be processed.

---

Table A4: Chain of Thought Prompt

Answer: The causal effect of cancer on X-ray is that X-rays are often used to diagnose or detect cancer in different parts of the body, such as the bones, lungs, breasts, or kidneys123. X-rays use low doses of radiation to create pictures of the inside of the body and show the presence, size, and location of tumors. X-rays can also help monitor the response to treatment or check for signs of recurrence. Therefore, having cancer may increase the likelihood of getting an X-ray as part of the diagnostic process or follow-up care. Therefore answer is <Answer>B</Answer>

Question: For a causal graph used to model relationship of various factors and outcomes related to cancer with the following nodes: ['Pollution', 'Cancer', 'Smoker', 'Xray', 'Dyspnoea'], Which cause-and-effect relationship is more likely between nodes 'pollution' and 'cancer'?

A. changing the state of node 'pollution' causally effects a change in another node 'cancer'.

B. changing the state of node 'cancer' causally effects a change in another node 'pollution'.

C. There is no causal relation between the nodes 'pollution' and 'cancer'.

Make sure to first provide a grounded reasoning for your answer and then provide the answer in the following format: <Answer>A/B/C</Answer>. It is very important that you output the final Causal graph within the tags like <Answer>A/B/C</Answer> otherwise your answer will not be processed.

Answer: The causal effect of pollution on cancer is that air pollution contains carcinogens (cancer-causing substances) that may be absorbed into the body when inhaled and damage the DNA of cells. Another possible reasoning is that particulate matter (tiny dust-like particles) in air pollution may cause physical damage to the cells in the lungs, leading to inflammation and oxidative stress and eventually cell mutations. A third possible reasoning is that air pollution may create an inflamed environment in the lungs that encourages the proliferation of cells with existing cancer-driving mutations. These are some of the hypotheses that researchers have proposed to explain how air pollution may cause cancer, but more studies are needed to confirm them. Therefore answer is <Answer>A</Answer>

Question: For a causal graph used to model relationship of various factors and outcomes related to cancer with the following nodes: ['Pollution', 'Cancer', 'Smoker', 'Xray', 'Dyspnoea'], Which cause-and-effect relationship is more likely between nodes 'pollution' and 'smoker'?

A. changing the state of node 'pollution' causally effects a change in another node 'smoker'.

B. changing the state of node 'smoker' causally effects a change in another node 'pollution'.

C. There is no causal relation between the nodes 'pollution' and 'smoker'.

Make sure to first provide a grounded reasoning for your answer and then provide the answer in the following format: <Answer>A/B/C</Answer>. It is very important that you output the final Causal graph within the tags like <Answer>A/B/C</Answer> otherwise your answer will not be processed.

Answer: Reason behind the lack of causal relation between pollution and smoker is that pollution and smoking are both independent risk factors for respiratory problems, but they do not directly cause or affect each other. Pollution and smoking both contribute to air pollution, which can harm the health of people and the environment. However, pollution is mainly caused by human activities such as burning fossil fuels, deforestation, or industrial processes, while smoking is a personal choice that involves inhaling tobacco smoke. Therefore, pollution and smoker are not causally related, but they are both associated with respiratory problems. Therefore answer is <Answer>C</Answer>.

Table A5: Chain of Thought Prompt (continued..)

Question: For a causal graph used for modeling factors causing Coronary Heart Diseases with the following nodes: ['Family Disease', 'Gene', 'Smoking', 'Blood Pressure', 'Coronary Heart Disease', 'Headache'], Which cause-and-effect relationship is more likely between nodes 'Family Disease' and 'Gene'?

A. changing the state of node 'Family Disease' causally effects a change in another node 'Gene'.

B. changing the state of node 'Gene' causally effects a change in another node 'Family Disease'.

C. There is no causal relation between the nodes 'Family Disease' and 'Gene'.

Make sure to first provide a grounded reasoning for your answer and then provide the answer in the following format: <Answer>A/B/C</Answer>. It is very important that you output the final Causal graph within the tags like <Answer>A/B/C</Answer> otherwise your answer will not be processed.

Answer: Reason behind the causal effect of family disease on gene is that family disease is a term that refers to diseases or health conditions that run in the family, meaning that they are influenced by genetic factors. Gene is a term that refers to the basic unit of heredity that carries information for a specific trait or function. Family disease can affect gene by altering the type or frequency of genes that are inherited by the offspring from their parents. For example, some family diseases are caused by deterministic genes, which are genes that guarantee the development of a disease if they are present in a person's genome. Other family diseases are influenced by risk genes, which are genes that increase the likelihood of developing a disease but do not guarantee it. Therefore, family disease can causally direct to gene by changing the genetic makeup of a person or a population. Therefore answer is <Answer>A</Answer>.

Question: For a causal graph used for modeling factors causing Coronary Heart Diseases with the following nodes: ['Family Disease', 'Gene', 'Smoking', 'Blood Pressure', 'Coronary Heart Disease', 'Headache'], Which cause-and-effect relationship is more likely between nodes 'Coronary Heart Disease' and 'Gene'?

 A. changing the state of node 'Coronary Heart Disease' causally effects a change in another node 'Gene'.

B. changing the state of node 'Gene' causally effects a change in another node 'Coronary Heart Disease'.

C. There is no causal relation between the nodes 'Coronary Heart Disease' and 'Gene'.

Make sure to first provide a grounded reasoning for your answer and then provide the answer in the following format: <Answer>A/B/C</Answer>. It is very important that you output the final Causal graph within the tags like <Answer>A/B/C</Answer> otherwise your answer will not be processed.

Answer: Possible reasoning behind the causal effect of gene on coronary heart disease is that gene is a term that refers to the basic unit of heredity that carries information for a specific trait or function. Gene can affect coronary heart disease by influencing the structure and function of the blood vessels, the metabolism and transport of lipids (fats) in the blood, the inflammation and clotting processes, or the response to environmental factors such as smoking or diet. For example, some genes code for proteins that regulate the cell cycle and growth of the cells that line the arteries, which can affect their susceptibility to damage or plaque formation. Other genes code for proteins that control the synthesis and clearance of cholesterol or other lipids, which can affect their levels and deposition in the arteries. Therefore, gene can causally direct to coronary heart disease by modifying the biological pathways that contribute to the development or progression of the disease. Therefore answer is <Answer>B</Answer>

Table A6: Chain of Thought Prompt (continued..)

Question: For a causal graph used for modeling factors causing Coronary Heart Diseases with the following nodes: ['Family Disease', 'Gene', 'Smoking', 'Blood Pressure', 'Coronary Heart Disease', 'Headache'], Which cause-and-effect relationship is more likely between nodes 'Blood Pressure' and 'Smoking'?

A. changing the state of node 'Blood Pressure' causally effects a change in another node 'Smoking'.

B. changing the state of node 'Smoking' causally effects a change in another node 'Blood Pressure'.

C. There is no causal relation between the nodes 'Blood Pressure' and 'Smoking'.

Make sure to first provide a grounded reasoning for your answer and then provide the answer in the following format: <Answer>A/B/C</Answer>. It is very important that you output the final Causal graph within the tags like <Answer>A/B/C</Answer> otherwise your answer will not be processed.

Answer: Possible reasoning behind the causal effect of smoking on blood pressure is that smoking is a habit that involves inhaling tobacco smoke, which contains nicotine and other harmful chemicals. Smoking can affect blood pressure by activating the sympathetic nervous system (SNS), which is the part of the nervous system that controls the body's response to stress or danger. When the SNS is activated, it releases hormones such as adrenaline and noradrenaline, which cause the heart to beat faster and harder, and the blood vessels to constrict. This results in a temporary increase in blood pressure, which can last for 15 to 20 minutes after each cigarette. Therefore, smoking can causally direct to blood pressure by stimulating the SNS and increasing the cardiac output and vascular resistance. Therefore answer is <Answer>B</Answer>.

Question: For a causal graph used for modeling factors causing Coronary Heart Diseases with the following nodes: ['Family Disease', 'Gene', 'Smoking', 'Blood Pressure', 'Coronary Heart Disease', 'Headache'], Which cause-and-effect relationship is more likely between nodes 'Headache' and 'Smoking'?

A. changing the state of node 'Headache' causally effects a change in another node 'Smoking'.

B. changing the state of node 'Smoking' causally effects a change in another node 'Headache'.

C. There is no causal relation between the nodes 'Headache' and 'Smoking'.

Make sure to first provide a grounded reasoning for your answer and then provide the answer in the following format: <Answer>A/B/C</Answer>. It is very important that you output the final Causal graph within the tags like <Answer>A/B/C</Answer> otherwise your answer will not be processed.

Answer: One possible reasoning behind the lack of causal relation between headache and smoking is that headache and smoking are both associated with various health conditions, but they do not directly cause or affect each other12. Headache is a term that refers to pain or discomfort in the head, scalp, or neck, which can have many possible causes, such as stress, dehydration, infection, injury, or medication. Smoking is a habit that involves inhaling tobacco smoke, which contains nicotine and other harmful chemicals, which can increase the risk of diseases such as cancer, heart disease, stroke, and lung disease. Therefore, headache and smoking are not causally related, but they are both linked to different health problems. Therefore the answer is <Answer>C</Answer>

Table A7: Chain of Thought Prompt (continued..)

Question: For a causal graph used for modeling factors causing Coronary Heart Diseases with the following nodes: ['Family Disease', 'Gene', 'Smoking', 'Blood Pressure', 'Coronary Heart Disease', 'Headache'], Which cause-and-effect relationship is more likely between nodes 'Headache' and 'Smoking'?

A. changing the state of node 'Headache' causally effects a change in another node 'Smoking'.

B. changing the state of node 'Smoking' causally effects a change in another node 'Headache'.

C. There is no causal relation between the nodes 'Headache' and 'Smoking'.

Make sure to first provide a grounded reasoning for your answer and then provide the answer in the following format: <Answer>A/B/C</Answer>. It is very important that you output the final Causal graph within the tags like <Answer>A/B/C</Answer> otherwise your answer will not be processed.

Answer: One possible reasoning behind the lack of causal relation between headache and smoking is that headache and smoking are both associated with various health conditions, but they do not directly cause or affect each other. Headache is a term that refers to pain or discomfort in the head, scalp, or neck, which can have many possible causes, such as stress, dehydration, infection, injury, or medication. Smoking is a habit that involves inhaling tobacco smoke, which contains nicotine and other harmful chemicals, which can increase the risk of diseases such as cancer, heart disease, stroke, and lung disease. Therefore, headache and smoking are not causally related, but they are both linked to different health problems. Therefore the answer is <Answer>C</Answer>

Question: For a causal graph used for modeling factors causing Coronary Heart Diseases with the following nodes: ['Family Disease', 'Gene', 'Smoking', 'Blood Pressure', 'Coronary Heart Disease', 'Headache'], Which cause-and-effect relationship is more likely between nodes 'Coronary Heart Disease' and 'Smoking'?

A. changing the state of node 'Smoking' causally effects a change in another node 'Coronary Heart Disease'.

B. changing the state of node 'Coronary Heart Disease' causally effects a change in another node 'Smoking'.

C. There is no causal relation between the nodes 'Coronary Heart Disease' and 'Smoking'.

Make sure to first provide a grounded reasoning for your answer and then provide the answer in the following format: <Answer>A/B/C</Answer>. It is very important that you output the final Causal graph within the tags like <Answer>A/B/C</Answer> otherwise your answer will not be processed.

Answer: Possible reasoning behind the causal effect of smoking on coronary heart disease is smoking damages the heart and blood vessels by raising triglycerides, lowering HDL, increasing blood clotting, and impairing blood flow to the heart. This can lead to plaque buildup, heart attacks, and death. Therefore answer is <Answer>A</Answer>.

Question: For a causal graph used for context with the following nodes: nodes, Which cause-and-effect relationship is more likely between nodes X and Y?

A. changing the state of node X causally effects a change in another node Y.

B. changing the state of node Y causally effects a change in another node X.

C. There is no causal relation between the nodes X and Y.

Make sure to first provide a grounded reasoning for your answer and then provide the answer in the following format: <Answer>A/B/C</Answer>. It is very important that you output the final Causal graph within the tags like <Answer>A/B/C</Answer> otherwise your answer will not be processed.

Table A8: Chain of Thought Prompt (continued..)

Which cause-and-effect relationship is more likely?

A. changing the state of node which says X causally effects a change in another node which says Y.

B. changing the state of node which says Y causally effects a change in another node which says X.

C. There is no causal relationship between node X and Y.

Make sure to first output a factually grounded reasoning for your answer. X and Y are nodes of a Causal Graph. The causal graph is sparse and acyclic in nature. So option C could be chosen if there is some uncertainity about causal relationship between X and Y.

First give your reasoning and after that please make sure to provide your final answer within the tags <Answer>A/B/C</Answer>.
It is very important that you output your final answer between the tags like <Answer>A/B/C</Answer> otherwise your response will not be processed.

Table A9: Base prompt

For the nodes X and Y which form an edge in a Causal Graph, you have to identify which cause-and-effect relationship is more likely between the nodes of the edge. This will be used to rearrange the nodes in the edge to create a directed edge which accounts for causal relation from one node to another in the edge.

A. changing the state of node X causally affects a change in another node Y.

B. changing the state of node Y causally affects a change in another node X.

C. There is no causal relation between the nodes X and Y.

You can also take the edges from the skeleton which have been rearranged to create a directed edge to account for causal relationship between the nodes: directed_edges.
Make sure to first output a factually grounded reasoning for your answer. First give your reasoning and after that please make sure to provide your final answer within the tags <Answer>A/B/C</Answer>.
It is very important that you output your final answer between the tags like <Answer>A/B/C</Answer> otherwise your response will not be processed.

Table A10: Iterative orientation prompt

For the following undirected edge in a Causal Graph made of nodes X and Y, you have to identify which cause-and-effect relationship is more likely between the nodes of the edge. This will be used to rearrange the nodes in the edge to create a directed edge which accounts for causal relation from one node to another in the edge.

A. changing the state of node X causally effects a change in another node Y.

B. changing the state of node Y causally effects a change in another node X.

C. There is no causal relation between the nodes X and Y.

You can also take the other directed edges of nodes X: X_edges and Y: Y_edges of the Causal graph as context to redirect the edge to account for causal effect.

Make sure to first output a factually grounded reasoning for your answer. First give your reasoning and after that please make sure to provide your final answer within the tags <Answer>A/B/C</Answer>.

It is very important that you output your final answer between the tags like <Answer>A/B/C</Answer> otherwise your response will not be processed.

Table A11: Markov Blanket prompt

*Identify the causal relationships between the given variables and create a directed acyclic graph to {context}. Make sure to give a reasoning for your answer and then output the directed graph in the form of a list of tuples, where each tuple is a directed edge. The desired output should be in the following form: [('A','B'), ('B','C')] where first tuple represents a directed edge from Node 'A' to Node 'B', second tuple represents a directed edge from Node 'B' to Node 'C' and so on.*

*If a node should not form any causal relationship with other nodes, then you can add it as an isolated node of the graph by adding it seperately. For example, if 'C' should be an isolated node in a graph with nodes 'A', 'B', 'C', then the final DAG representation should be like [('A','B'), ('C')].*
*Use the description about the node provided with the nodes in brackets to form a better decision about the causal direction orientation between the nodes.*

*It is very important that you output the final Causal graph within the tags <Answer></Answer>otherwise your answer will not be processed.*

*Example:*
*Input: Nodes: ['A', 'B', 'C', 'D'];*
*Description of Nodes: [(description of Node A), (description of Node B), (description of Node C), (description of Node D)]*
*Output: <Answer>[('A','B'),('C','D')]</Answer>*
*Question:*
*Input: Nodes: {Triplet Nodes Input}*
*Description of Nodes: {Description of Each Node from the Triplet}*
*Output:*

Table A12: The *triplet* prompt template, which includes a concise context of the graph, the input triplet nodes and their respective descriptions. As an example, for the Child graph, the context is *"to model congenital heart disease in babies"*, the three nodes may be *('HypoxiaInO2', 'Grunting', 'GruntingReport')*; and their node descriptions are *["hypoxia when breathing oxygen", "grunting in infants", "report of infant grunting"]* respectively.

---

Input: **('HypDistrib', 'LowerBodyO2')**

Answer: Low oxygen areas equally distributed around the body can affect the level of oxygen in the lower body by reducing the amount of oxygen available for circulation. Therefore, the answer is <Answer>A</Answer>.

Input: **('Disease', 'LungFlow')**

Answer: Infant methemoglobinemia is a condition where the blood is unable to carry enough oxygen, which can lead to low blood flow in the lungs. Therefore, the answer is <Answer>A</Answer>.

Input: **('CardiacMixing', 'HypDistrib')**

Answer: Mixing of oxygenated and deoxygenated blood is a common cause of congenital heart disease in babies. This mixing can occur due to structural defects in the heart, such as a hole between the heart chambers or abnormal connections between the blood vessels. This mixing can lead to low oxygen areas equally distributed around the body. Therefore, the answer is <Answer>B</Answer>.

---

Table A13: Example LLM (GPT-3.5-turbo) outputs for estimating the causal direction between a given pair of nodes (with their descriptions) using the CoT Prompt (refer Table A4 in Appendix for the prompt). Given a pair (X,Y), Option $A$ refers to X causing Y, option $B$ refers to Y causing X, and option $C$ refers to no causal relationship between the variables. The pairs shown are from the *Child* graph from BNLearn repository (Scutari & Denis, 2014), that is used for modeling Congenital Heart Disease in infants.

---

Input: **('Right C7','Discoligment injury C6-C7')**

Answer: Discoligment injury C6-C7 can cause compression of the nerve roots that exit the spinal cord at the C7 level, which can lead to symptoms such as pain, numbness, and weakness in the right C7 dermatome. Therefore, the answer is <Answer>B</Answer>.

Input: **('Right C7', 'Left C7')**

Answer: Right C7 and left C7 are both parts of the cervical spine and are not known to directly influence each other. Therefore, the answer is <Answer>C</Answer>.

Input: **('Right elbow trouble', 'Left shoulder trouble')**

Answer: There is no direct causal relationship between right elbow trouble and left shoulder trouble. They may both be symptoms of a larger underlying condition, but they do not directly cause or affect each other. Therefore the answer is <Answer>C</Answer>.

---

Table A14: Example LLM (GPT-3.5-turbo) reasoning outputs for estimating causal directionality between different pairs of nodes using CoT Prompt (refer Table A4 for the prompt) for Neuropathic subgraph (used for pain diagnosis).

| Dataset | Samples | LLM | Ground Truth | PC (Average over MEC) |
|---------|---------|-----|--------------|----------------------|
| Asia | 250 | 1.00±0.00 | 0.00±0.00 | 2.00±0.00 |
| | 1000 | 3.00±0.00 | 2.00±0.00 | 3.00±0.00 |
| | 10000 | 3.00±0.00 | 3.00±0.00 | 3.00±0.00 |
| Child | 250 | 5.00±0.00 | 5.00±0.00 | 6.50±0.00 |
| | 1000 | 6.00±0.00 | 6.00±0.00 | 8.43±0.00 |
| | 10000 | 9.00±0.00 | 9.00±0.00 | 9.75±0.00 |

Table A15: Comparing $D_{top}$ of final graph using LLM order vs Ground truth order as prior to PC algorithm for Child and Asia graph, averaged over 4 runs

| Dataset | Samples | $\epsilon_{ATE}(S_1)$ | $\epsilon_{ATE}(S_2)$ | $\epsilon_{ATE}(S_3)$ | $\Delta_{12}$ | $\Delta_{13}$ |
|---------|---------|----------------------|----------------------|----------------------|---------------|---------------|
| Asia | 250 | 0.70±0.40 | 0.70±0.39 | 0.69±0.39 | 0.00±0.00 | 0.00±0.00 |
| | 500 | 0.64±0.39 | 0.64±0.39 | 0.64±0.38 | 0.00±0.00 | 0.00±0.00 |
| | 1000 | 0.59±0.32 | 0.59±0.32 | 0.59±0.32 | 0.00±0.00 | 0.00±0.00 |
| | 5000 | 0.59±0.30 | 0.59±0.30 | 0.59±0.29 | 0.00±0.00 | 0.00±0.00 |
| | 10000 | 0.49±0.00 | 0.49±0.00 | 0.49±0.00 | 0.00±0.00 | 0.00±0.00 |

Table A16: Results on Asia dataset. Here we test the difference in the estimated causal effect of *lung* on *dyspnoea* when the causal effect is estimated using the backdoor set $S_1$ = *{smoke}* vs. the causal effect estimated when all variables in two topological orders as backdoor sets: $S_2$ = *{asia, smoke}*, $S_2$= *{asia, tub, smoke}*. $\Delta_{12}, \Delta_{13}$ refers to the absolute difference between the pairs $\epsilon_{ATE}(S_1), \epsilon_{ATE}(S_2)$ and $\epsilon_{ATE}(S_1), \epsilon_{ATE}(S_3)$ respectively. From the last two columns, we observe that using the variables that come before the treatment node in a topological order as a backdoor set does not result in the deviation of causal effects from the ground truth effects.

