# OpenReview forum: "Causal Inference Using LLM-Guided Discovery"
_ICLR.cc/2024/Conference — Submitted to ICLR 2024_

### Official Review · Reviewer_2Neo · 2023-10-27

**Soundness:** 2 fair
**Presentation:** 3 good
**Contribution:** 2 fair
**Rating:** 3
**Confidence:** 4

**Summary:**

The authors propose to use LLMs with majority voting to learn a causal order of the random variables in the underlying data generating process represented by directed acyclic graphs from observed data. The learned causal order is then used to orient the undirected edges in the output of the existing causal discovery algorithms. Additionally, the authors claim that causal graphs are not necessary needed for causal effects estimation, rather, the causal order is sufficient by finding a valid backdoor adjustment set. They further argue that using causal orders is preferable in the case when domain expert knowledge is available.

**Strengths:**

The authors demonstrate the utility of LLMs in causal discovery through means of causal orders and use that as a background knowledge for the existing causal discovery algorithms.  The paper also shows that causal structures are not necessarily required for causal effect estimation and causal orders are sufficient. It also shows both empirically and theoretically that SHD is not a good metric to measure the accuracy of predicting correct causal orders. The paper is fairly well-written and the proofs are sound.

**Weaknesses:**

* Taking outputs from LLMs as inputs to causal discovery algorithms is not uncommon [5]. I find the comparison in the experiment is not quite fair to the existing causal discovery algorithms. There are many existing algorithms that incorporate background knowledge of ordering restrictions [1, 2, 3] and they are not reported on the paper. The authors could have randomly sampled from the ground truth and provided that as background knowledge to other algorithms in the experiment especially for graphs that are less than 20 nodes to compare against methods with LLMs. Given that the theoretical contributions are relatively small, I would expect to see more empirical experiments to show the strong motivation and merits of the approach.

* The experimental result could have been highly affected by the popularity of the datasets and domain knowledge on the internet and using LLMs to guide causal discovery can be very limited to those commonly available data.

* It is not clear what the advantages of using LLMs as a source of domain knowledge are as it may have issues with hallucinations unless there are large-scale experiments that show some domain knowledge are impractical to obtain via domain experts and need LLMs to guide such effort.

* It is also not clear to me why the estimation is not compared against with those estimation methods that use causal graphs or simply a Markov equivalence class of DAGs [4] even if there is only the information of causal orders available to show the merits of using only the causal order for estimation.

References

* [1] de Campos, Luis M., and Javier G. Castellano. "Bayesian network learning algorithms using structural restrictions." International Journal of Approximate Reasoning 45.2 (2007): 233-254.
* [2] Cooper, Gregory F., and Edward Herskovits. "A Bayesian method for the induction of probabilistic networks from data." Machine learning 9 (1992): 309-347.
* [3] Borboudakis, Giorgos, and Ioannis Tsamardinos. "Towards robust and versatile causal discovery for business applications." Proceedings of the 22nd ACM SIGKDD International Conference on Knowledge Discovery and Data Mining. 2016
* [4] Jung, Yonghan, Jin Tian, and Elias Bareinboim. "Estimating identifiable causal effects on markov equivalence class through double machine learning." International Conference on Machine Learning. PMLR, 2021.
* [5] Taiyu Ban, Lyvzhou Chen, Xiangyu Wang, and Huanhuan Chen. From query tools to causal architects: Harnessing large language models for advanced causal discovery from data. arXiv preprint
arXiv:2306.16902, 2023.

**Questions:**

1. How does using triples helps avoiding cycles in learning the causal order?
2. Is it possible that the causal order output by LLMs orient a new unshielded collider in the output of other causal discovery algorithms?
3. Have the authors tried to provide background knowledge to PC and compare that with PC+LLM? For example, randomly sample from the ground truth and provide such background knowledge to PC or other algorithms.

---

> ### Author Response · Authors · 2023-11-22
> **Response to Third Reviewer**
>
> We thank the reviewer for their valuable feedback. We have incorporated their feedback and provided a review wise explanation as follows:
>
> **Response To Weaknesses**
>
> **Comment**: Taking outputs from LLMs as inputs to causal discovery algorithms is not uncommon [5] ..... show the strong motivation and merits of the approach.
>
> **Response**:  Our work focuses more on evaluating how good LLMs are as knowledge bases or domain experts rather than focusing on how well different discovery algorithms incorporate domain knowledge. The work leverages LLMs understanding of real world, to extract relevant information (causal order) which is inputted into causal discovery algorithms as priors to further aid their performance. While we agree past work has covered inputting LLM outputs as priors to discovery algorithms, our work stands out by focusing on probing LLMs for extracting causal order instead of detecting all possible edges, due to causal order’s utility in downstream tasks like effect estimation, prediction, etc. While previous work builds upon pairwise prompting strategies our work proposes a novel triplet based method which overcomes the previous methods drawbacks of having cycles in the final graph and presenting a lower DTop.
>
> To present an analysis of how close LLMs as knowledge bases are to ground truth information as priors, we conduct an experiment on Child and Asia dataset (refer Table A15).
>
> **Comment**: The experimental result could have been highly...limited to those commonly available data.
>
> **Response**: We agree that there is a possibility that LLMs have seen the BNLearn datasets in its pretraining setup, and highlight this in the limitations section of our paper as well. However to tackle this claim we include the Neuropathic Pain Diagnosis dataset in our updated paper draft and show the effectiveness of our methods on the same. The dataset is less popular and requires a very nuanced medical understanding in order to correctly assess the causal relationship, therefore depicting how LLMs can be employed in real world settings which are not straightforward. That being said, we still agree that memorization of causal discovery datasets is an issue in current evaluation of LLMs which is scope for future research work.
>
> **Comment**: It is not clear what the advantages...need LLMs to guide such effort.
>
> **Response**: One of the practical implications of using LLMs can be to aid human experts as they can look at the LLM generated graphs and edit it further thus saving time as well as efforts. While LLMs do suffer from issues like hallucinations, they still carry an understanding of the world which can be used to assist the human experts to improve and fasten the causal graph creation process. It will be on human experts to accept the suggestions and reasoning provided by LLMs while constructing causal graphs thus removing any hallucinated or false outcome.
>
> **Comment**: It is also not clear to me why the estimation...the causal order for estimation.
>
> **Response**: Thank you for the review, we have added a comparative analysis (Refer Table A16 in Appendix) on Asia dataset to show estimation using backdoor set from the given graph vs given causal/topological order of the graphs.
>
> **Response To Questions**
>
> **Question**: How does using triples helps avoiding cycles in learning the causal order?
>
> **Answer**: Our intuition is that due to dynamic context, i.e. with each iteration for deciding edge wise causal direction between a given pair of nodes, the third variable changes thus providing different context each time. Due to this, the LLM gets an overall understanding of the other nodes present in the graph. Thus, the aggregate of all the decisions LLM makes pertaining to a specific pair followed by tie breaker using GPT-4, incorporates overall understanding plus robustness in the final answer. Higher incorporation of neighboring nodes plus multiple querying of while deciding causal edge between all possible pairs, might be some reasons behind lower number of cycles. On the other hand, since pairwise analysis that has been done in previous literature, prompt LLM only once without extra contextual information, the edge formations are not always robust and might be contributing factors for forming more cycles.
>
> **Question**: Is it possible that the causal order output by LLMs orient a new unshielded collider in the output of other causal discovery algorithms?
>
> **Answer**: Since we do not have a check for this in our current algorithm, there is a slight possibility that LLMs might orient new unshielded colliders when used as priors with PC Algorithm. We will incorporate this check in our pipeline.
>
> **Question**: Have the authors tried to provide background knowledge to PC...knowledge to PC or other algorithms.
>
> **Answer**:  We have added Table A15 in Appendix to show comparison between LLM and Ground truth prior for causal discovery.

---

### Official Review · Reviewer_cXcH · 2023-10-30

**Soundness:** 4 excellent
**Presentation:** 3 good
**Contribution:** 2 fair
**Rating:** 5
**Confidence:** 3

**Summary:**

The paper addresses the question if and how LLMs can be utilized for causal discovery tasks. For this, the authors focus on effect estimation and argue that knowledge about the causal order is sufficient. The paper aims at two contributions: 1) Showing that the causal order is sufficient for effect estimation problems and 2) showing how LLMs can be used in addition to statistical approaches, such as PC, to improve the causal discovery performance. The suggested approach has been evaluated with different experiments.

**Strengths:**

- The paper addresses a logical step to combine causal discovery approaches with the domain 'knowledge' of LLMs.
- Careful consideration of different approaches on using LLMs.
- Encouraging results in the experiments.

**Weaknesses:**

The overall idea is a logical next step seeing the recent success of LLMs in the causal context. However, some of my concerns are:
- The first contribution regarding the sufficiency about knowing the causal order is not novel and a rather straightforward insight seeing that conditioning on any 'upstream' node of a treatment variable in a DAG results in a valid adjustment set. Therefore, it is certainly good to point this out again, but this is not a new contribution by this work.
- The paper overall seems rather incremental, seeing that the paper by Kiciman et al. is already providing some significant prior work in this regard for causal discovery. However, I acknowledge the incorporation of LLM generated knowledge with statistical approaches such as PC.

See the "Questions" section for further points.

**Questions:**

My main concern is the rather incremental novelty, especially since the argument that the causal order is sufficient for effect estimation tasks is a well known point. Some other remarks:

- You are focusing on effect estimation tasks, but the general premise of using LLMs for causal discovery can also be helpful for other tasks. Consider formulating it more broadly and then focus only on effect estimation in the experiments as an example.
- You are arguing that looking at SHD is often the wrong metric. However, these works using SHD typically address the problem of inferring the whole DAG structure without any particular causal task in mind, while you are only concerned with the causal order for effect estimation problems. In that sense, the SHD makes sense as a metric to see how good the inferred DAG structure is.
- While you reference the work by Kiciman et al., a more direct comparison is missing. In particular, the related open source package https://github.com/py-why/pywhy-llm has several prompting techniques for inferring structural information. That being said, they do not combine it with methods like PC, which is the novel part in your work.
- Fair discussion of the limitations and potential issues with overfitting.

---

> ### Author Response · Authors · 2023-11-22
> **Response to Second Reviewer**
>
> We thank the reviewer for their valuable inputs, we have provided review wise explanation as follows:
>
> **Weaknesses**
> **Comment**: The first contribution regarding the sufficiency....but this is not a new contribution by this work.
> **Response**: The technical result may not be a new result for the backdoor criterion. But its application leads to a valid order-based method for causal discovery using LLMs, whereas we argue that existing pairwise-based prompts that elicit edges may not be valid. The pre-existing work has heavily leveraged LLMs for causal edge detection between any given nodes by optimizing on metrics like SHD for full graph discovery. But the identification of direct causal edge between any given pair of nodes heavily depends on the presence of other nodes, therefore we believe probing LLMs for getting the order between any given pair is a better question framing since the pairwise order only depends on the variables in the question.
>
> For example, consider the data-generating process, lung cancer ->doctor visit -> Positive Xray. If asked, an expert would affirm a causal edge from ‘Lung cancer’ to ‘Positive Xray’ (indeed, such an edge exists in the BNLearn Cancer [1]). However, if they are told that the set of observed variables additionally includes doctor visit, then the correct answer would be to not create a direct edge between ‘Lung Cancer’ and ‘Positive Xray’, but rather create edges mediated through ‘doctor visit’. Note that the causal order, ‘Lung Cancer’ < ‘Positive Xray’ remains invariant in both settings. To fill this conceptual gap in the existing work, our work focuses on causal order and probes LLMs specifically from this perspective since the causal order is useful for most downstream tasks and in practice has huge applications as it is locally consistent in terms of estimation (refer table 1).
>
> **Comment**: The paper overall seems rather incremental...with statistical approaches such as PC.
>
> **Response**: While our work draws inspiration from prior research to enhance LLM-based pipelines for aiding in causal discovery, our approach stands out by prioritizing the exploration of how LLMs can effectively identify causal order. This emphasis on identifying causal order is more valuable compared to solely focusing on direct and indirect edge connections, which often rely heavily on other variables. But since order between any given pair of variables remains invariant to other present variables and they directly correlate with downstream tasks like Causal effect estimation, we emphasize and build our frameworks towards that. While previous papers have focused on pairwise causal relationship discovery, our work takes inspiration from pre-existing algorithms to propose novel LLM based pipelines (such as the Triplet prompt). Also, we present how LLMs can be easily adapted with different classes of discovery algorithms whether it is score based or constraint based, for enhancing discovery performance. Most importantly, we emphasize on how current LLMs can be used for downstream applications like causal inference, effect estimation, prediction, etc by leveraging them as knowledge base for finding correct causal order whereas previous work (like Kiciman et al.) have focused more on pairwise causal relationship identification for graph discovery. We show how this approach is not suitable and does not leverage LLMs potential to help in causal tasks by helping get the correct order of the graph by aiding discovery algorithms like PC, CamML, etc.

---

> > ### Author Response · Authors · 2023-11-22
> > **Response to Second Reviewer (Questions)**
> >
> > **Response to Questions**
> >
> > **Q**: My main concern is the rather incremental novelty ... is a well known point.
> > **A**: While we do agree that causal order is sufficient for effect estimation, our work provides the novelty in how we are utilizing LLMs as domain experts in association with different classes of discovery algorithms. While pre-existing LLM pipelines have focused on pairwise evaluation of all node pairs for graph discovery, we bring in our own Triplet based methods for correct order estimation to optimize LLMs for downstream tasks and use it for causal discovery with algorithms like CamML and PC.
> >
> > **Q**: You are arguing that looking at SHD is often the wrong metric. However ... how good the inferred DAG structure is.
> >
> > **A**: While we do agree that SHD is a good metric for evaluating similarity of two graphs, applying this metric specifically for evaluating LLMs’ performance for causal discovery is something we disagree with since direct and indirect edge connections cannot be determined due to the subjectivity of the task and it’s dependence on other variables present (as demonstrated in the ‘Lung Cancer’, ’Positive X-Ray’ and ‘Doctor Visit’ example above from Cancer dataset). While consideration of other variables can have a direct impact on edge formation between any given pair of nodes, the causal order remains invariant. Therefore probing LLMs to find order is more valid, and also transfers to downstream causal tasks like effect estimation, prediction etc. which further helps position LLMs for utility in real world applications.
> >
> > **Q**: While you reference the work by Kiciman et al., a more direct comparison is missing. In particular, the related open source package https://github.com/py-why/pywhy-llm has several prompting techniques for inferring structural information. That being said, they do not combine it with methods like PC, which is the novel part in your work.
> >
> > **A**: Thank you for pointing this out, the base prompt which we have used in our work has been taken directly from Kiciman et al. We further compare this with various prompting strategies including our Triplet prompt for a more direct comparative analysis on DTop and SHD. We have updated the manuscript and cited Kiciman et al. for the Base prompt.
> >
> > References: [1] M. Scutari and J.B. Denis. Bayesian Networks: With Examples in R Chapman & Hall/CRC Texts in Statistical Science. Taylor & Francis, 2014

---

### Official Review · Reviewer_vBFn · 2023-11-01

**Soundness:** 3 good
**Presentation:** 2 fair
**Contribution:** 1 poor
**Rating:** 3
**Confidence:** 3

**Summary:**

As a method for estimating causal effects, this paper proposes using LLMs as virtual experts to elicit a causal ordering of the variables. With the causal ordering, a valid backdoor set can be determined as the causal effect can be estimated. Different prompting strategies are explored, as well as algorithms that combine these virtual expert judgments with existing causal discovery algorithms.

UPDATE: I appreciate the authors' reply, which alleviates my concerns about soundness. However, is still think this paper's contribution is weak, so my overall assessment remains unchanged.

**Strengths:**

The results are presented fairly well.

Replacing human experts by LLMs could be considered, though I am not up-to-date on the related work cited for this part of the paper.

**Weaknesses:**

The theoretical contribution is trivial, and contains multiple mistakes.

**Questions:**

* Assumption 3.3 states there is no latent confounding between treatment and target, but you actually need the stronger assumption that there is no latent confounding between any observed variables. Otherwise for instance proposition 4.2 will fail: Suppose we want to find a valid backdoor set for $X \to Y$, and there is a third observed variable $Z$ that is not a cause or effect of $X$ or $Y$, but there is a latent variable causing $X$ and $Z$, and another causing $Y$ and $Z$. Then a valid topological ordering of the observed variables is $Z < X < Y$, but adjusting for $Z$ actually opens the backdoor path.

* Proposition 4.2 requires the further assumption that $i < j$.

* Paragraph below proposition 4.2, "causal effect practitioners tend to include all confounders ...": Can you provide a reference for this claim? Either way, what you propose goes further than including all *confounders*: you also include variables that cause either the target or the treatment but not both.

* The definitions of $E_m, E_f, $E_d$ for SHD are incorrect: a wrongly oriented edge will add one to each of these three variables. Further, I think you mean to add the cardinalities rather than the sets themselves.

* Algorithm in section 5.2: Steps 2 and 3 and the difference between them are unclear from the text. For algorithms, it may be better to use pseudocode, or at least some mathematical notation.

* In the prompts in the appendix, I noticed that often "causally effects" is written when "causally affects" was meant.

---

> ### Author Response · Authors · 2023-11-22
> **Response to Reviewer 1**
>
> We thank the reviewer for their valuable inputs, we have incorporated their comments and provided a review wise explanation as follows:
>
> **Question**: Assumption 3.3 states there is no latent confounding....observed variables is Z<X<Y, but adjusting for Z actually opens the backdoor path.
>
> **Answer**: Thank you for pointing it out and we agree with you. We revised the manuscript to include the line “We assume that the underlying causal graph has no unobserved confounders''.
>
>
> **Question**: Proposition 4.2 requires the further assumption that i<j.
>
> **Answer**: Thank you for pointing it out. We’ve updated the proposition statement in the revised manuscript.
>
>
> **Question**: Paragraph below proposition 4.2, "causal effect practitioners tend to include all confounders ...":....the treatment but not both.
>
> **Answer**: The statement is based on causal inference practice in statistics and econometrics where all observed covariates (assumed to be pre-treatment) are included in the estimation model. As noted by Cinelli and Pearl (2020), variables that cause only the target are “good” variables to condition on, so they do not lead to any trouble. Including causes of treatment only can lead to estimation issues (e.g., including an instrumental variable) that increase with the strength of the instrument’s causal effect on treatment. In practice, however, strong instruments are rare. https://papers.ssrn.com/sol3/papers.cfm?abstract_id=3689437
>
>
> **Question**: The definitions of Em, Ef, E_d$ for SHD are incorrect: a wrongly oriented edge will add one to each of these three variables. Further, I think you mean to add the cardinalities rather than the sets themselves.
>
> **Answer**: Apologies. We’ve removed the mathematical formula for SHD as it is not required to understand the rest of the paper and the detailed formula will affect the readability.
>
>
> **Question**: Algorithm in section 5.2: Steps 2 and 3 and the difference between them are unclear from the text. For algorithms, it may be better to use pseudocode, or at least some mathematical notation.
>
> **Answer**: We’ve updated the Algorithm 2 in the revised manuscript to clearly explain the steps.
>
>
> **Question**: In the prompts in the appendix, I noticed that often "causally effects" is written when "causally affects" was meant.
>
> **Answer**: We apologize for the typographical mistake, we have updated in the main manuscript

---

### Meta-Review · Area_Chair_Z9xj · 2023-12-04

**Metareview:**

This paper proposes a LLM-Guided method for estimating causal effects, by using LLMs as virtual experts to elicit a causal ordering of the variables. With the causal ordering, a valid backdoor set can be determined as the causal effect can be estimated.

A reasonable amount of discussions took place between the authors and the reviewers. In the end, we got three reviews with ratings of 3, 5, and 3 with confidence of 3, 3, and 4 respectively.

The experiment result is interesting (vBFn, cXcH, 2Neo). However, all the reviewers (vBFn, cXcH, 2Neo) suggest that the contribution is limited. The decision is reject.

**Justification For Why Not Higher Score:**

The issues raised by reviewers should be properly addressed especially the concerns of theoretical contributions.

**Justification For Why Not Lower Score:**

N/A

---

### Decision · Program_Chairs · 2024-01-16

Reject